# Tuning polymer-backbone coplanarity and conformational order to achieve high-performance printed all-polymer solar cells

Yilei Wu ®[1], Yue Yuan[2], Diego Sorbelli ®[3], Christina Cheng[4], Lukas Michalek ®[1], Hao-Wen Cheng[1], Vishal Jindal ®[5], Song Zhang[1], Garrett LeCroy[4], Enrique D. Gomez ®[6], Scott T. Milner[5], Alberto Salleo ®[4], Giulia Galli ®[3], John B. Asbury[2], Michael F. Toney ®[7] & Zhenan Bao ®[1] ✉

All-polymer solar cells (all-PSCs) offer improved morphological and mechanical stability compared with those containing small-molecule-acceptors (SMAs). They can be processed with a broader range of conditions, making them desirable for printing techniques. In this study, we report a high-performance polymer acceptor design based on bithiazole linker (PY-BTz) that are on par with SMAs. We demonstrate that bithiazole induces a more coplanar and ordered conformation compared to bithiophene due to the synergistic effect of non-covalent backbone planarization and reduced steric encumbrances. As a result, PY-BTz shows a significantly higher efficiency of 16.4% in comparison to the polymer acceptors based on commonly used thiophene-based linkers (i.e., PY-2T, 9.8%). Detailed analyses reveal that this improvement is associated with enhanced conjugation along the backbone and closer interchain π-stacking, resulting in higher charge mobilities, suppressed charge recombination, and reduced energetic disorder. Remarkably, an efficiency of 14.7% is realized for all-PSCs that are solution-sheared in ambient conditions, which is among the highest for devices prepared under conditions relevant to scalable printing techniques. This work uncovers a strategy for promoting backbone conjugation and planarization in emerging polymer acceptors that can lead to superior all-PSCs.

Polymer solar cells (PSCs) have attracted considerable interest as an unconventional renewable energy source due to several unique benefits such as low-cost solution processing, light weight and semi-transparency properties, and mechanical flexibility[1,2]. The state-of-the-art PSCs based on wide-bandgap polymer donor and narrow-bandgap nonfullerene small-molecule acceptors (NF-SMAs) have achieved power conversion efficiencies (PCEs) over 18%[3–8]. However, to realize commercial products and applications, high performance must also be accompanied by scalable processing conditions and excellent stability[9–12]. Despite the high efficiency of PSCs based on SMAs, the excessive sensitivity to the film coating conditions, the low glass transition temperatures ($T_g$) and fast diffusion of these SM-NFAs leads

[1]Department of Chemical Engineering, Stanford University, Stanford, CA 94305-4125, USA. [2]Department of Chemistry, The Pennsylvania State University, University Park, PA 16802, USA. [3]Pritzker School of Molecular Engineering, University of Chicago, 5747 South Ellis Avenue, Chicago, IL 60637, USA. [4]Department of Materials Science and Engineering, Stanford University, Stanford, CA 94305, USA. [5]Department of Chemical Engineering, The Pennsylvania State University, University Park, PA 16802, USA. [6]Department of Chemical Engineering and Department of Materials Science and Engineering, The Pennsylvania State University, University Park, PA 16802, USA. [7]Department of Chemical and Biological Engineering, Materials Science Program, Renewable and Sustainable Energy Institute, University of Colorado Boulder, Boulder, CO 80309, USA. ✉e-mail: zbao@stanford.edu

to poor scalability and unstable blend morphologies within the device active layer. Compared with the SMAs-based PSCs, all-polymer solar cells (all-PSCs) comprising both polymer donor and polymer acceptor have superior morphological, thermal, and mechanical stability, as well as better compatibility with large-scale film coating techniques, which make them more desirable for durable and scalable applications[13–18]. On the other hand, the PCEs of all-PSCs still lag behind those of the SMAs-based PSCs due to the lack of available high-performance polymer acceptors and challenges in controlling morphology[19,20]. Therefore, more efforts are needed to further develop all-PSCs.

Some of the most extensively studied polymer acceptor for all-PSCs are based on perylenediimide and naphthalenediimide building blocks[17,21–24]. However, these acceptors suffer from poor light absorption efficiency and short-lived exciton lifetimes[17,21–24]. To solve these issues, a new strategy, namely polymerized small molecule acceptor (PSMA), was reported by Li and coworkers, where a narrow-bandgap fused-aromatic-ring type SMA that serves as a main core unit is copolymerized with a π-bridge linker (i.e., thiophene derivatives)[25]. Recently, all-PSCs based on the PSMAs have demonstrated PCE of over 15%, making their performance closer to SMA-based PSCs[26–28]. However, the efficiency of all-PSCs has not reached its full potential, which is largely due to the relatively low FF resulted from challenging morphology control and typically poor electron transport. Achieving optimal microstructure is indeed a major challenging task for the polymer-polymer blend systems because the large molecular weight of polymers leads to negligible entropic gain upon mixing, which can significantly reduce the overall driving force for polymer miscibility[29].

To improve the device performance of such all-PSCs, researchers have largely focused on optimizing SMA building blocks[30,31], while small consideration has been paid to the π-bridge linkers[32–34]. The reported PSMAs typically used unsubstituted thiophene, selenophene[35] or BDT[36,37] as π-bridge linkers. We hypothesize that π-bridge linking units can have significant effect on the exciton separation, charge transport, charge recombination, and charge extraction processes through tuning of conjugated polymer conformations and chain packing. Specifically, we speculate on the possibility of introducing weak non-covalent bonding interactions[38–44] into the π-linker to stabilize planar conformations and minimize conformational disorder. In complex systems such as π-conjugated macromolecules, a large

conformational space is possible. Therefore, reducing disorder with nonbonding interactions is a desirable direction because the optoelectronic properties of these materials depend strongly on the molecular conformation and the packing structure of the polymer chains. Thus, engineering of the π-linker to tune the polymer conformations is a promising strategy to further increase the PCE of the all-PSCs through rational design.

In this work, we investigate the effect of π-linker units on the polymer conformation, and its correlation with performance of all-PSCs. We designed and synthesized a new electron acceptor, namely PY-BTz as shown in Fig. 1, composed of A-DAD-A type building block and 4,4′-dialkoxy-5,5′-bithiazole (BTz) as the π-linker. PY-BTz possesses significantly enhanced electron mobility and an intrinsically low energetic disorder relative to the commonly used oligothiophene-linked PSMAs (i.e., PY-2T, Fig. 1). This is due to better polymer backbone co-planarity, resulting in a better intra-molecular conjugation and stronger inter-chain packing, as supported by DFT modeling, measured electronic properties, and X-ray diffraction. In comparison to bithiophene (2T), BTz is a promising π-bridge linking unit due to its unique geometric and electronic properties: (1) the S(thiazolyl)···O(alkoxy) attraction[44] promotes a non-covalent conformational lock with extensive π-conjugation and high molecular order; (2) swapping (thiophene)C–H with (thiazole)N decreases steric hindrance by removing repulsive C–H···H–C interactions between neighboring arene units, thereby further improving π–conjugation, π–stacking and packing order; and (3) electron-deficiency nature of thiazole counteracts alkoxy electron-donating characteristics, which lowers the highest occupied molecular orbital (HOMO) energy of the acceptor polymer and provides a better air stability. In addition, we systematically investigate the effect of donor polymer with different degrees of fluorination (PBDB-TFx, with x = 0, 0.25, 0.50, 0.75, and 1, Fig. 1 and Supplementary Fig. 1) on the blend morphology and the photovoltaic performance of all-PSCs. The PBDB-TF$_{0.25}$ leads to the optimal polymer blend miscibility and energy-level matching. Consequently, the PBDB-TF$_{0.25}$:PY-BTz based devices exhibit the highest PCE of 16.4% (PCE$_{ave}$ = 16.0 ± 0.2%), which is significantly higher than that of PY-2T (9.8 ± 0.1%). Moreover, a high PCE of 14.7 (PCE$_{ave}$ = 13.9 ± 0.6%) can be achieved in devices fabricated using solution shearing under ambient conditions, and long-time stability of devices was observed.

**Fig. 1 | Synthesis routes of PY-BTz and PY-2T.** Compared with PY-2T, PY-BTz is designed to have a more co-planar and rigid polymer backbone due to (1) the S(thiazolyl)···O(alkoxy) attraction promoting a non-covalent conformational lock effect and (2) the replacement of (thiophene)C–H with (thiazole)N removing repulsive C–H···H–C interactions with neighboring arene units.

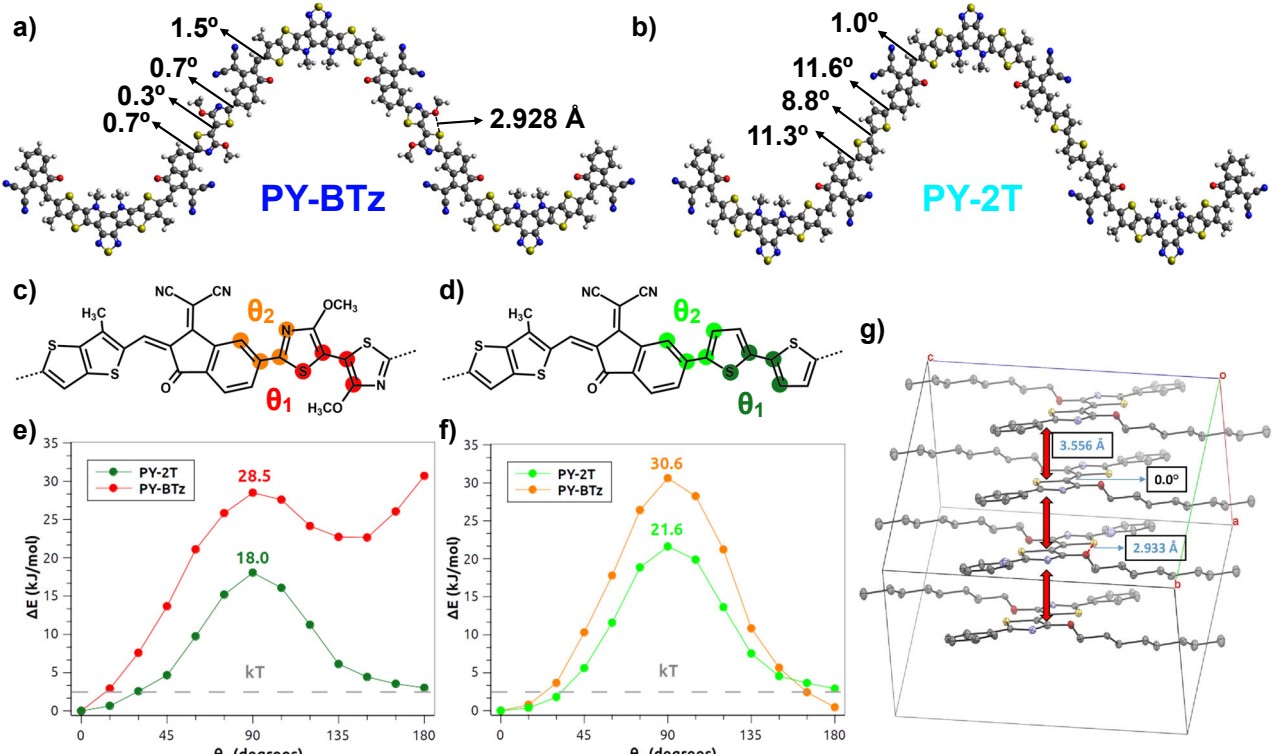

**Fig. 2 | Molecular structures analysis.** Optimized neutral ground-state trimer structures of (**a**) PY-BTz and (**b**) PY-2T calculated by DFT at the B3LYP-D3 level of theory. Red: oxygen atoms; yellow: sulfur atoms; blue: nitrogen atoms. The most relevant torsion angles (with acute angle notation) and bond lengths are reported. Schematic structure of central TT-IC-BTz (**c**) and TT-IC-2T (**d**) units of PY-BTz and PY-2T, respectively, together with the representation of the torsion angles describing the rotation around T(z)-T(z) ($\theta_1$) and IC-T(z) ($\theta_2$) bonds. Potential energy surface scans along dihedral angles $\theta_1$ (**e**) and $\theta_2$ (**f**) for PY-BTz and PY-2T model dimers. The gray dashed lines indicate thermal energy at 298 K. **g** Single-crystal XRD showing the planar conformation of model compound phenyl-BTz.

This PCE is among the highest for solar cells made under conditions applicable to scalable printing techniques. Our work shows that the unique structural and electronic properties of BTz make it a promising π-bridge linking unit for polymer acceptors and demonstrates that conjugated polymers with more planar and rigid backbone conformation through π-linker optimization and non-covalent bonding interactions can simultaneously achieve high electron mobility, good polymer-blend miscibility, and minimal performance loss for printed OPVs.

## Results

### Synthesis

The detailed synthetic protocols and characterizations of the polymers are shown in the Supplementary Information. Briefly, by co-polymerizing Y6-OD-2Br (Fig. 1) with 4,4'-Bis(octyloxy)-2,2'-bis(-trimethylstannyl)-5,5'-bithiazole or 5,5'-bis(trimethylstannyl)-2,2'bithiophene via Stille polycondensation, two PSMAs, PY-BTz or PY-2T were obtained, respectively. The two compounds have comparable average molecular weights ($M_n$) around 13 kDa and dispersity (Đ) <2.8, which minimize the effects of molecular weight on the polymer properties. Their chemical structures were further characterized by matrix-assisted laser desorption ionization time of flight mass spectrometry (MALDI-TOF-MS, Supplementary Fig. 2), Fourier-transform infrared spectroscopy (FT-IR, Supplementary Fig. 3), and proton nuclear magnetic spectroscopy (1H NMR, Supplementary Figs. 4–9). Both polymers are readily dissolved in organic solvents like chlorobenzene and 1,2-dichlorobenzene. They also show similar good thermal stability ($T_d$ at 5% weight loss under nitrogen atmosphere at 372 °C and 375 °C, respectively), as determined by thermogravimetric analysis (TGA, Supplementary Fig. 10). We did not observe any thermal transition for these polymers from differential scanning calorimetry

(DSC) data in the range of −80 to 350 °C. Higher molecular-weight polymers (over 20 kDa) show significantly lower solubility leading to more difficult thin film processing and are therefore not included in this study.

### DFT calculations

To show the impact of linker units on the structural and electronic properties of the polymers, we performed density functional theory (DFT) calculations using hybrid functionals and dispersion interactions (at the B3LYP-D3 level of theory; see Computational Details) (Fig. 2). It is apparent from the structures of the fully relaxed trimer conformations in Fig. 2a, b that the dihedral angle formed between the SMA and the π-bridging unit ($\theta_2$) is very small (nearly 1°) for PY-BTz, while the same angles increases to over 11° in the case PY-2T, due to the removal of the steric C−H···H−C repulsions with neighboring arene units. Similarly, within the linker units, the dihedral angle between two thiazoles ($\theta_1$) is also reduced (0.3°) compared to that between two thiophenes (8.8°) for the two trimer conformers, which is attributed to intramolecular noncovalent S(thiazolyl)···O(alkoxy) attraction within the BTz[45,46]. To further address this behavior, we computed the potential energy surface scan along $\theta_1$ and $\theta_2$ for model PY-BTz and PY-2T dimers and the results suggest a similar trend of conformations. As a matter of fact, the potential energy surfaces along these angles have a steeper slope and a higher associated rotational barrier for PY-2T, indicating a more planar conformation for PY-BTz (Fig. 2e, f). This result is further corroborated computationally by a denser-grid potential energy surface scan of the same torsional angles for model systems TT-IC-BTz and TT-IC-2T (Supplementary Fig. 11) and experimentally by the single-crystal XRD analysis of model compound Ph-BTz (Fig. 2g and Supplementary Data 1 and 2). Furthermore, our computational optimization of coiled

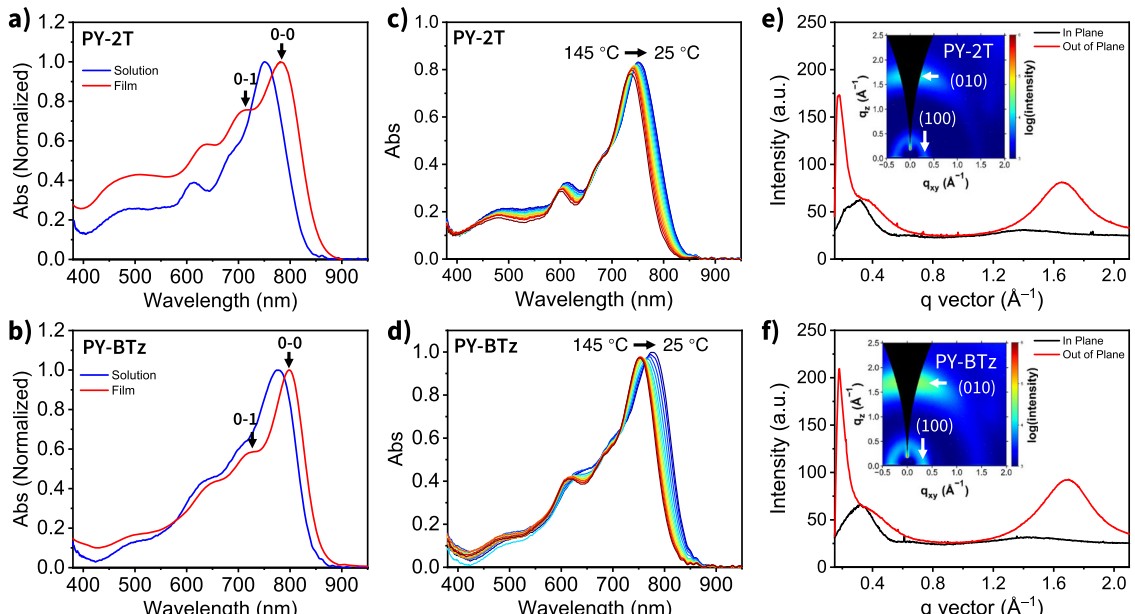

**Fig. 3 | Optical and morphological properties of neat polymers.** Normalized UV–Vis–NIR absorption spectra of (**a**) PY-2T and (**b**) PY-BTz in trichlorobenzene solution and in thin-film state. Variable-temperature UV–Vis–NIR absorption spectra of (**c**) PY-2T and (**d**) PY-BTz in solution from 145 to 25 °C. Plots of 2D GIWAXS for the (**e**) PY-2T and (**f**) PY-BTz thin films and corresponding in-plane and out-of-plane scattering profiles.

conformers of PY-2T and PY-BTz trimers (Supplementary Fig. 12) resulted in a highly twisted structure for PY-2T, close in energy to that of the zig-zag conformer shown in Fig. 2b (5.8 kJ/mol); we also find that the coiled PY-BTz conformer exhibits less twisting and lies at a much higher energy (44.1 kJ/mol). These results demonstrate that the structural changes identified in our calculations lead to better backbone coplanarization and π-conjugation for PY-BTz and thus lead to a more delocalized lowest unoccupied molecular orbital (LUMO) electron cloud distribution, stronger electron coupling, and smaller optical bandgap (*vide infra*). In fact, the intra-chain tight-binding hopping matrix elements[47], calculated from the LUMO−LUMO + 1 energy difference[48] in model dimer compounds using DFT, are 44.5 and 29.1 meV for PY-BTz and PY-2T, respectively. The larger hopping matrix element across the linkers for the BTz system corresponds to higher electron coupling, hopping rates and mobilities, which are indeed consistent with experimental observations (*vide infra*). Additionally, sp² hybridized N atoms in the bithiazole unit are expected to induce electron-deficiency at the linker site for PY-BTz. Both Molecular Electrostatic Potential (MEP) maps of PY-2T and PY-BTz monomers (Supplementary Fig. 13) and the related MEP-derived atomic charges (Supplementary Fig. 14) are fully consistent with this hypothesis. Such deficiency is expected to favor negative charge delocalization in PY-BTz and, in turn, to mitigate exciton recombination losses in devices, consistent with experimental results (*vide infra*). Overall, our calculations reveal that the PY-BTz has less polymer backbone twisting, which is desirable for better intermolecular packing and electron transport, and contains more electron-deficient units, which are expected to reduce recombination losses.

## The optical and morphological properties of the polymer acceptors

The room-temperature trichlorobenzene solution UV–Vis–NIR absorption peaks ($\lambda_{max,sol}$) of PY-BTz and PY-2T are found (Fig. 3a, b) at 778 and 754 nm, respectively. This red shift of absorption band is indicative of the better intramolecular conjugation through planar BTz linker that leads to increased persistence lengths and/or a stronger tendency for aggregation. In thin-films state, PY-BTz exhibits an absorption peak ($\lambda_{max, film}$) at 799 nm, while that of reference PY-2T is

located at 780 nm. Correspondingly, the photoluminescence (PL) spectrum is red shifted (Supplementary Fig. 15). Thus, PY-BTz has a smaller optical bandgap ($E_g$) while having comparable HOMO energy as shown by photoelectron spectroscopy in air (PESA, Supplementary Fig. 16). Moreover, the absorbance ratio between the 0−0 and 0−1 vibronic peak is higher for PY-BTz ($A_{0-0}/A_{0-1} = 1.71$) than for PY-2T ($A_{0-0}/A_{0-1} = 1.32$), which could be attributed to the better inter-molecular packing and stronger electronic coupling. To better understand the aggregation behavior difference of these two compounds, we measured their variable-temperature UV–Vis–NIR absorption spectra in dilute trichlorobenzene solution (0.02 mg/ml) from 145 to 25 °C (Fig. 3c, d). Upon the decrease in temperature, the absorption peak maximum redshifts as consequence of backbone planarization and aggregation. Importantly, PY-BTz can maintain a consistently more red-shifted spectra than that of PY-2T at all temperatures, implying that its polymer backbone is more rigid under the same physical condition and exhibits less disaggregation upon temperature increase[49].

Next, we carried out grazing incidence wide angle X-ray scattering (GIWAXS) measurements to understand the effect of π-linker on polymer ordering and crystallinity in neat films. The relevant crystallographic parameters are summarized in Table 1. Both polymer thin film samples show a preferential face-on orientation, with a lamellar stacking peak (100) in in-plane (IP) direction and a π-π stacking peak (010) in the out-of-plane (OOP) direction (Fig. 3e, f), which is known to facilitate vertical charge transport in all-PSC[50]. Importantly, the change of π-linker from bithiophene in PY-2T to bithiazole in PY-BTz decreases π-stacking distance ($d_{010}$) from 3.80 Å to 3.70 Å, and enhances the corresponding coherence length ($CCL_{010}$) from 10 Å to 13 Å. The combined, DFT, UV–Vis–NIR absorption, and GIWAXS results suggest that the π-spacer facilitates polymer self-assembly both in solution and in the thin-film state.

The different polymer aggregation and crystallinity properties correlates with difference in charge transport properties, as shown by the electron mobility ($\mu_e$) measurements by using the space-charge-limited current (SCLC) method (Supplementary Fig. 17a and Table 1). The $\mu_e$ of PY-BTz ($5.4 \pm 0.9 \times 10^{-4}$ cm² V⁻¹ s⁻¹) is indeed higher than that of PY-2T ($2.1 \pm 0.3 \times 10^{-4}$ cm² V⁻¹ s⁻¹), suggesting that the BTz based

**Table 1 | Physicochemical properties of neat polymer acceptors[a]**

| Polymer | $\lambda_{max}$ [nm] (solution) | $\lambda_{max}$ [nm] (film) | $E_g$[b] [eV] | LUMO/HOMO[b] [eV] | $q_{010}$ [Å⁻¹] | $d_{010}$ [Å] | $\mu_e$ [$10^{-4}$ cm² V⁻¹ s⁻¹] |
|---|---|---|---|---|---|---|---|
| PY-2T | 754 | 780 | 1.45 | −4.21/−5.66 | 1.65 | 3.80 ± 0.1 | 2.1 ± 0.3 |
| PY-BTz | 778 | 799 | 1.42 | −4.24/−5.66 | 1.70 | 3.70 ± 0.1 | 5.4 ± 0.9 |

[a]Averages from at 3 samples.

[b]HOMO energy estimated from photo-electron spectroscopy in air (PESA). LUMO energy calculated as HOMO + optical band gap $E_g$ (calculated from the absorption onset of the films).

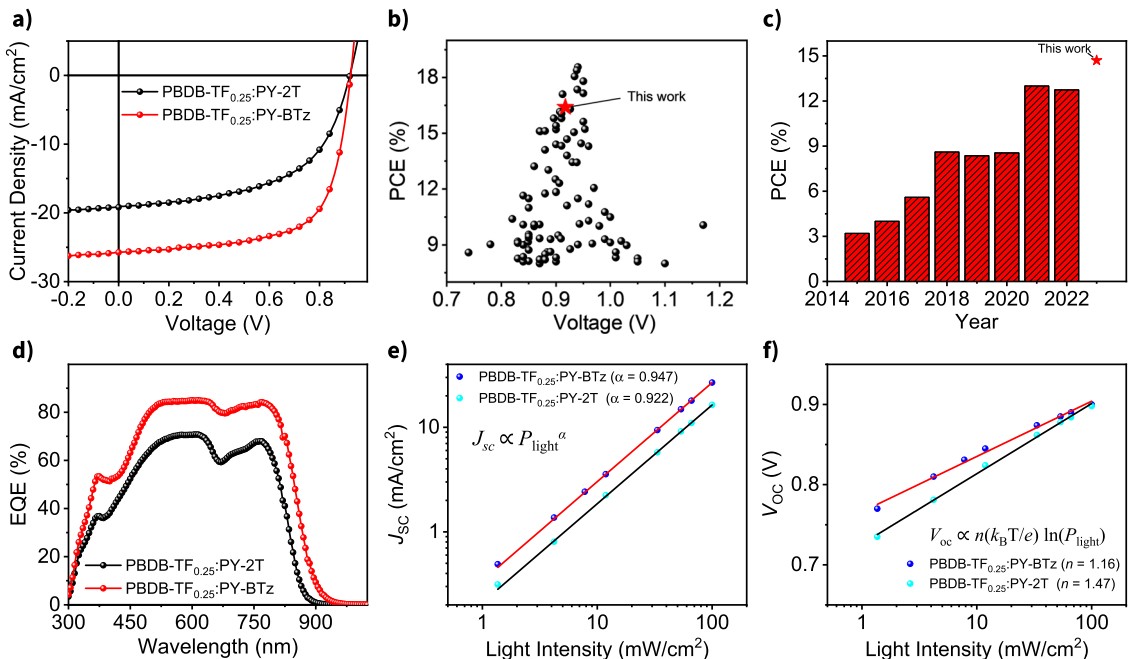

**Fig. 4 | Device performance of all-PSCs. a** *J−V* characteristics of PY-2T (blank) and PY-BTz (red) based all-PSCs devices. **b** Plot of PCE against $V_{OC}$ for reported spin-coated all-PSCs (for the references, Supplementary Table 4). **c** Highest-performing PCE values of solution printed all-PSCs for each year since 2015 (for the references, see Supplementary Table 5). **d** EQE spectra of PY-2T (blank) and PY-BTz (red) based all-PSCs devices. Light intensity dependence of (**e**) $J_{SC}$ and (**f**) $V_{OC}$ for PY-2T and PY-BTz based all-PSCs devices.

π-bridge effectively enhanced electron mobility of the PSMAs due to aforementioned conformational and morphological properties.

**The photovoltaic properties of all-PSCs**

Next, we made all-PSCs with an inverted device structure of ITO/ZnO/PFNBr/PBDB-TFx:PSMAs/MoO₃/Ag to characterize the photovoltaic parameters. The optimization procedures are detailed in the Supplementary Information. Briefly, PBDB-TF$_{0.25}$ was found to be the best polymer donor (Supplementary Fig. 18), likely due to a compromise between optimal energy-level matching and blend miscibility. The current density versus voltage (*J−V*) curves of the optimized PBDB-TF$_{0.25}$:PY-BTz and PBDB-TF$_{0.25}$:PY-2T devices are depicted in Fig. 4a, and their photovoltaic characteristics are summarized in Table 2 and in Supplementary Fig. 19. Compared to the devices based on PY-2T acceptor, the PY-BTz-based devices produce a comparable open-circuit voltage ($V_{OC}$), but a larger $J_{SC}$ of 25.8 mA cm⁻², together with a greater FF of 67%. Thus, the PBDB-TF$_{0.25}$:PY-BTz devices show a higher PCE of up to 16.4% (PCE$_{ave}$ of 16.0 ± 0.2%, Table 2), outperforming those of the PBDB-TF$_{0.25}$:PY-2T system (9.77 ± 0.07%). This PCE value is among the highest to date for spin-coating processed all-PSCs (Fig. 4b).

Given that transitioning from small-area device to large-area device while retaining high PCE is challenging, we proceeded to test these new compounds processed by blade-coating in ambient. The PBDB-TF$_{0.25}$:PY-BTz system with the best performance was selected to prepare blade-coated devices. For blade coating, the ink formulation is kept the same as that used in spin-coating, while the coating speed is

optimized to achieve the optimal film thickness of 105 ± 5 nm. Remarkably, the all-PSCs device based on PBDB-TF$_{0.25}$:PY-BTz fabricated by facile blade-coating achieved a champion PCE of 14.7% (PCE$_{ave}$ = 13.9 ± 0.6%, Table 2), which is, to the best of our knowledge, the highest PCE to date for all-PSCs prepared by blade-coating in ambient condition (Fig. 4c). This result indicates that in the PBDB-TF$_{0.25}$:PY-BTz-based system, the solar cell performance difference between spin-coating and blade-coating is small, which could be attributed to the low sensitivity of the all-polymer blend morphology to processing conditions. Moreover, the comparison of thermal stability of the unencapsulated all-PSCs is tested in a nitrogen-filled glovebox baked at 90 °C is shown in Supplementary Fig. 20. After 175 h, 92.4% ± 1.9% of the initial PCE was maintained by the optimal blade-coated devices, which was higher than the 85.1% ± 1.4% for the spin-coated device. We speculate that this improvement in thermal stability might be due to relatively slower film drying kinetics of blade-coating method compared to spin-coating, resulting in fewer kinetic traps or instabilities in the blend film morphology. Further studies are currently underway to corroborate this hypothesis.

**Charge generation, recombination, and transport properties**

We measured the exciton dissociation and charge recombination characteristics in the active layers to further understand the factors contributing to the higher performance of PY-BTz based all-PSCs when compared to devices made from PY-2T. To elucidate the $J_{SC}$ change, external quantum efficiency (EQE) measurements of the

**Table 2 | The photovoltaic parameters[a] of the optimized All-PSC devices based on PBDB-TF$_{0.25}$:PY-BTz and PBDB-TF$_{0.25}$:PY-2T under the illumination of AM 1.5 G, 100 mW cm$^{-2}$**

| Acceptor | Method | $J_{sc}$ (mA/cm²) | $V_{OC}$ (V) | Fill factor | PCE$_{ave}$ (%) | PCE$_{max}$ (%) | $\mu_h/\mu_e$ [$10^{-4}$ cm² V$^{-1}$ s$^{-1}$] |
|---|---|---|---|---|---|---|---|
| PY-2T | Spin-coating | 19.00 ± 0.18 | 0.922 ± 0.003 | 0.559 ± 0.006 | 9.77 ± 0.07 | 9.85 | 3.5 ± 0.5/1.8 ± 0.2 |
| PY-BTz | Spin-coating | 25.8 ± 0.7 | 0.917 ± 0.005 | 0.67 ± 0.02 | 16.0 ± 0.2 | 16.4 | 4.2 ± 0.5/3.8 ± 0.4 |
| PY-BTz | Solution-shearing in ambient | 22.2 ± 1.0 | 0.901 ± 0.006 | 0.693 ± 0.007 | 13.9 ± 0.6 | 14.7 | n.d. |

[a]The average values with standard deviations were obtained 10 samples (for the error analysis see Supplementary Fig. 19). Device structure of ITO/ZnO/PFNBr/Active-layer/MoO$_3$/Ag. Device area: 4 mm².

representative devices were first carried out. As shown in Fig. 4d, the EQE spectra of both samples can approach 80–85% in the range of 470–650 nm, but the PBDB-TF$_{0.25}$:PY-BTz shows an expanded spectral response up to ≈900 nm. PBDB-TF$_{0.25}$:PY-2T shows a narrower EQE response (872 nm) likely because of the weaker conjugation and poorer morphology, which harms the $J_{SC}$ of the resulting all-PSC. The integrated $J_{SC}$ determined from the corresponding EQE response (14.9 mA cm$^{-2}$ for PBDB-TF$_{0.25}$:PY-BTz and 18.8 mA cm$^{-2}$ for PBDB-TF$_{0.25}$:PY-2T) are consistent with the $J_{SC}$ extracted from the $J$–$V$ curves.

Photoluminescence quenching efficiency (PLQE) analysis was performed to understand charge dissociation behavior of the all-PSCs (Supplementary Fig. 22). The two neat polymer acceptors exhibit comparable PL quantum yields. After blending with PBDB-TF$_{0.25}$, both the blend films show a very efficient PL quenching phenomenon. The PBDB-TF$_{0.25}$:PY-BTz exhibits a slightly more efficient exciton-dissociation process (PLQE ≈ 0.97), which would explain a larger $J_{SC}$ in the 700–850 nm region as shown in the EQE experiment[51]. Next, we measured the dependence of the photocurrent density ($J_{ph}$) on the effective voltage ($V_{eff}$) (Supplementary Fig. 23). The exciton dissociation probabilities ($P_{diss}$) can be estimated by the ratio of $J_{ph}$ to the saturation photocurrent density ($J_{sat}$) as previously reported[52–54]. The values of $J_{ph}/J_{sat}$ of the all-PSCs based on PBDB-TF$_{0.25}$:PY-2T and PBDB-TF$_{0.25}$:PY-BTz were calculated to be 90% and 95%, respectively, suggesting that PY-BTz-based system has relatively higher exciton dissociation probability, in good agreement with PLQE measurements. The dependence of $J_{sc}$ on the illumination light intensity ($P_{light}$) was also investigated to provide further insight into mechanisms for charge carrier recombination[55,56]. In general, the dependence of $J_{sc}$ on $P_{light}$ follows the equation $J_{sc} \propto P_{light}^{\alpha}$, where the value of α can be a signature of the type of recombination that dominates within the device. The plots of log $J_{sc}$ vs. log $P_{light}$ (Fig. 4e) show that the α values, given by the slope of the plots, are 0.92 for the PY-2T and 0.95 for the PY-BTz-based devices. The α value closer to 1 for the all-PSC based on PBDB-TF$_{0.25}$:PY-BTz implies that there is less bimolecular recombination losses in these devices. The $V_{OC}$ was also plotted against log($P_{light}$) with the slope of $n(k_B T/e)$ (Fig. 4f), where $n$ is the ideal factor, $k_B$ is the Boltzmann constant, $T$ is the temperature, and $e$ is the elementary charge. The PBDB-TF$_{0.25}$:PY-BTz blend afforded a smaller $n$ value of 1.16 than the PY-2T (1.47) based device, indicating more suppression of trap-assisted Shockley-Read-Hall (SRH) recombination inside the PY-BTz based all-PSCs. All these combined results assert that recombination losses within PBDB-TF$_{0.25}$:PY-BTz devices have been significantly diminished, which can partially explain its smaller voltage loss and higher PCE than in PBDB-TF$_{0.25}$:PY-2T. To elucidate the improved FF in PBDB-TF$_{0.25}$:PY-BTz, the hole ($\mu_h$) and electron ($\mu_e$) mobility were investigated by SCLC methodology (Supplementary Fig. 17b, c). The hole- and electron-only devices were made with the structure of ITO/PEDOT:PSS/active-layer/MoO$_3$/Ag and ITO/ZnO/active-layer/LiF/Al, respectively[57]. As presented in Table 2, the calculated $\mu_h/\mu_e$ values for PBDB-TF$_{0.25}$:PY-2T are 3.5 × 10$^{-4}$/1.8 × 10$^{-4}$ cm² V$^{-1}$ s$^{-1}$, which increase to 4.2 × 10$^{-4}$/3.8 × 10$^{-4}$ cm² V$^{-1}$ s$^{-1}$ for PBDB-TF$_{0.25}$:PY-BTz. Thus, owing to the better electron mobility, PBDB-TF$_{0.25}$:PY-BTz shows a more balanced $\mu_h/\mu_e$ ratio, which is

expected to facilitate the charge extraction and collection of the corresponding all-PSCs, leading to improved FF[32].

To further comprehend photoinduced charge transfer (CT) and recombination dynamics[58], we performed femtosecond transient absorption spectroscopy (fs-TA) on the PY-2T and PY-BTz based samples. The exciton spectrum of the pristine PY-BTz film (Supplementary Fig. 24, with a pump wavelength at 800 nm) consists of three negative ground state bleach (GSB) features at 700–800 nm, and a positive excited state absorption (ESA) feature at 550 nm. The CT behavior of the PBDB-T$_{0.25}$:PY-BTz blend, with selective excitation of PY-BTz in the blend at 800 nm, is displayed in Fig. 5a, b (and Supplementary Fig. 25). At early times within 0.5 ps, in addition to PY-BTz GSB, another negative signal appears at 550–630 nm concurrently. This is assigned to the GSB feature of the PBDB-T$_{0.25}$. This donor GSB feature is an indication of ultrafast hole transfer from the PY-BTz exciton to PBDB-TF$_{0.25}$, producing the CT state at the donor-acceptor interface. The CT state features decay, and the fs-TA spectrum is then dominated by a long-lived charge-separated (CS) state that mainly includes donor GSB feature at 630 nm and PY-BTz GSB feature at 800 nm. Then, the fs-TA spectra of PBDB-TF$_{0.25}$:PY-2T blend films were measured under identical experimental conditions. As shown in Fig. 5c, d, while the spectral profiles are similar for the two blends, their dynamics are very distinctive. Comparing the relative intensity of the donor GSB feature at 630 nm (Fig. 5b, d) shows that the CT state is more pronounced in the PY-BTz based blend than in the PY-2T blend. This indicates a higher photoinduced hole transfer yield in the PBDB-TF$_{0.25}$:PY-BTz based device, consistent with PLQE results and with the smaller phase separation domain sizes measured by morphology analysis (vide infra). Furthermore, the normalized donor GSB feature at 600 nm decays slower in the PY-BTz blend compared to that in the PY-2T blend (Fig. 5e), suggesting that the PBDB-TF$_{0.25}$:PY-BTz blend shows a relatively longer CT state lifetime. The observed longer CT state lifetime is evidence of significantly less geminate charge-recombination loss within the polymer blend (Fig. 5f), consistent with the significantly improved $J_{sc}$ and FF of the PBDB-TF$_{0.25}$:PY-BTz system. We attribute this crucial improvement to the better capability of the PY-BTz to delocalize the negative charge compared to PY-2T. At a later timescale, under the open voltage condition of the transient absorption experiment, the polaron decays with a predominant bimolecular recombination mechanism (i.e., a two-body free carrier mechanism) as indicated by the nearly linear relation of $\Delta A^{-1}$ with time and is further analyzed by the nanosecond transient absorption spectroscopy.

We proceed to understand the resulting long-lived charge separated state properties in the blend films using IR-based nanosecond transient absorption experiments (ns-TRIR)[59]. Since only free polarons produced in the blends have a long-lived broad absorption in the IR range, ns-TRIR has the advantage of selectively measuring the polaron decay kinetics within blend films. Figure 6a, b shows the log-log plot of unnormalized polaron decay kinetics traces observed at 1800 cm$^{-1}$ probe frequency in PY-BTz and PY-2T based blend films following light excitation at 532 nm with different pump pulse intensities. The broad polaron absorption spectrum of the PY-BTz

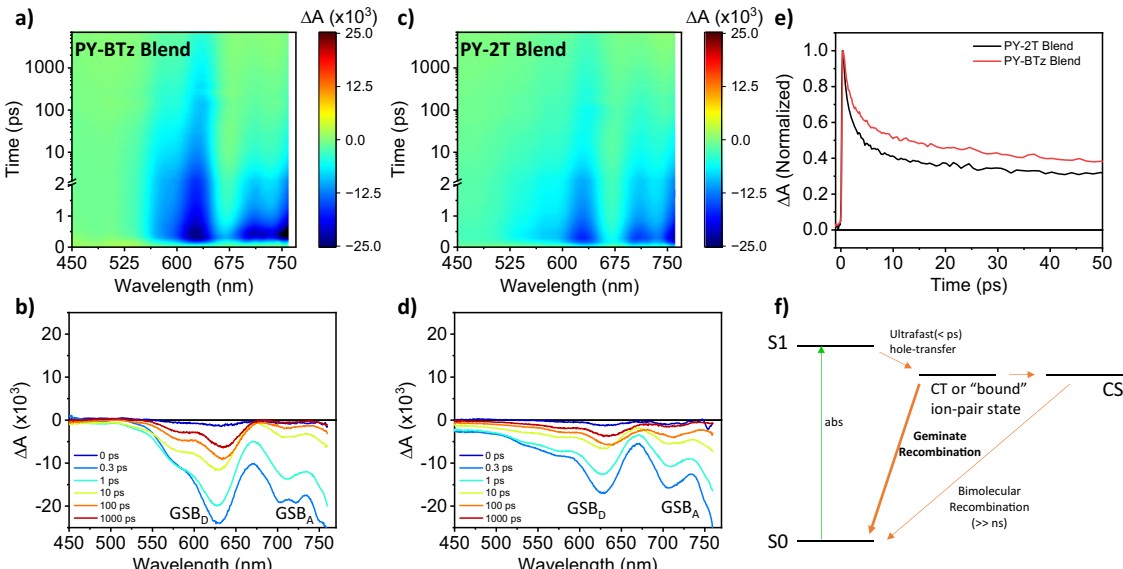

**Fig. 5 | Spectral and temporal resolved ultrafast exciton dissociation and charge recombination.** Femtosecond transient absorption spectra of (**a**, **b**) PY-BTz and (**c**, **d**) PY-2T based blend films by selectively exciting the polymer acceptor at 800 nm under same experimental conditions. **e** Normalized transient absorption decay kinetics at 600 nm showing slower CT state decay rate for PY-BTz based blend film (red line) respect PY-2T based one (black line). **f** Energy diagram illustrating the drop of CT state population through geminate recombination process.

blend film is included as an inset in Fig. 6a with a vertical line marking the 1800 cm$^{-1}$ probe frequency at which the kinetics were measured. The polaron absorption spectra measured in PY-2T blend films are similar and not shown. The PY-BTz based blend films clearly show higher polaron absorption intensity (Fig. 6c, average from 3 individual samples) across all excitation intensities. The polaron absorption intensities represent the average of multiple measurements of three samples to demonstrate reproducibility. All measurements exhibited the same monotonic growth of signal with increasing excitation intensity. The error bars reflect the sample-to-sample variation. While there may be subtle differences in the polaron absorption cross sections comparing the PY-BTz versus the PY-2T blend films, they are unlikely to account for the factor of two increase in polaron absorption with PY-BTz because of the similarity in structure and composition of the blend films. This finding is consistent with the slower geminate charge recombination observed in the fsTA results. The spectroscopic evidence of more charge carriers (polarons) present in the PY-BTz blend film is also consistent with its higher $J_{SC}$ and $V_{OC}$ values observed in the all-PSC device results (Fig. 4). This is because more charge carriers survive into the nanosecond to microsecond time scale in the PY-BTz blend film to be collected at electrodes of the devices. On the other hand, a plot of the $\alpha$ value describing the bimolecular power law decay (Fig. 6d) shows that PY-BTz and PY-2T based blend films have similar but modest $\alpha$ values. The $\alpha$ values are the result of averaging multiple measurements with the error bars indicating the sample-to-sample variation and reflect the depth of the trap state distribution for a particular organic semiconductor at a given temperature[60]. The data indicate comparable long-range charge transport properties of both PY-BTz and PY-2T based blend films. The observations of similar transport properties with significantly higher density of long-lived charge carriers in the PY-BTz blend film is particularly revealing because it points out the substantial role of the BTz based π-linker on reducing geminate recombination losses rather than bimolecular recombination processes in the all-PSCs.

### Blend film morphology characterization

To correlate the charge generation and transport properties with the blend film morphology, GIWAXS, Atomic force microscopy (AFM), resonant soft X-ray scattering (R-SoXS), and transmission electron microscopy (HRTEM) were carried out. GIWAXS measurements were first performed to study the microstructures and molecular packing behaviors of the polymer blend thin films (Fig. 7a, b)[61]. After blending with PBDB-TF$_{0.25}$, both PY-BTz and PY-2T based samples show predominant face-on orientation characterized by the π−π (010) stacking diffractions located at ca. 1.67 Å$^{-1}$ in the OOP direction (Fig. 7d) and the lamellar (100) diffractions at $q_z \approx 0.30$ Å$^{-1}$ in the IP direction (Fig. 7c). The face-on orientation is, therefore, dominant in the blends as in their neat films, which is advantageous for charge transport in diode configurations. Notably, PY-BTz based pristine and blend films both exhibit slightly larger crystal coherence lengths (CCL$_{100}$ and CCL$_{010}$) than the PY-2T systems, suggesting a relatively higher order of PY-BTz (Supplementary Fig. 26 and Supplementary Table 3), and contributing to better charge transport.

AFM imaging of the active layer was performed to characterize surface morphology. The PY-BTz-based blends exhibit a slightly larger root-mean-square roughness compared to PY-2T-based ones, which is beneficial for enlarging the contact area with electrodes and thus improves charge collection (Fig. 7e, f). More importantly, both the height and phase images (Fig. 7g, h) indicate that the PY-BTz-based blend films also possess slightly more well-defined and finer fibrillar morphology, which could contribute to better charge transport and fill factor. To further elucidate the nature of the detected domains, Derjaguin−Muller−Toporov (DMT) modulus[62] images were collected via PeakForce Quantitative Nanomechanical property Mapping (QNM; Fig. 7k, l and Supplementary Fig. 27). As both the observed domains and the surroundings show comparable DMT moduli (ca. 1 GPa), no major compositional differences are to be expected. Here, the AFM images clearly show a continuous interpenetrating network with a feature size in the range of 10−20 nm—a highly desirable morphology for good exciton dissociation and charge transport. Remarkably, a small reduction of the domain size is observed in PY-BTz-based blend films compared to PY-2T blends, giving rise to enhanced photovoltaic parameters (Table 2 and Fig. 4).

To measure changes in the internal structure of blend we performed transmission R-SoXS measurements. This method is well known to boost the contrast between two constituent materials and give details on the degree of nanoscale phase separation[63–67]. Figure 7m shows R-SoXS profiles of PBDB-TF$_{0.25}$:PY-BTz and

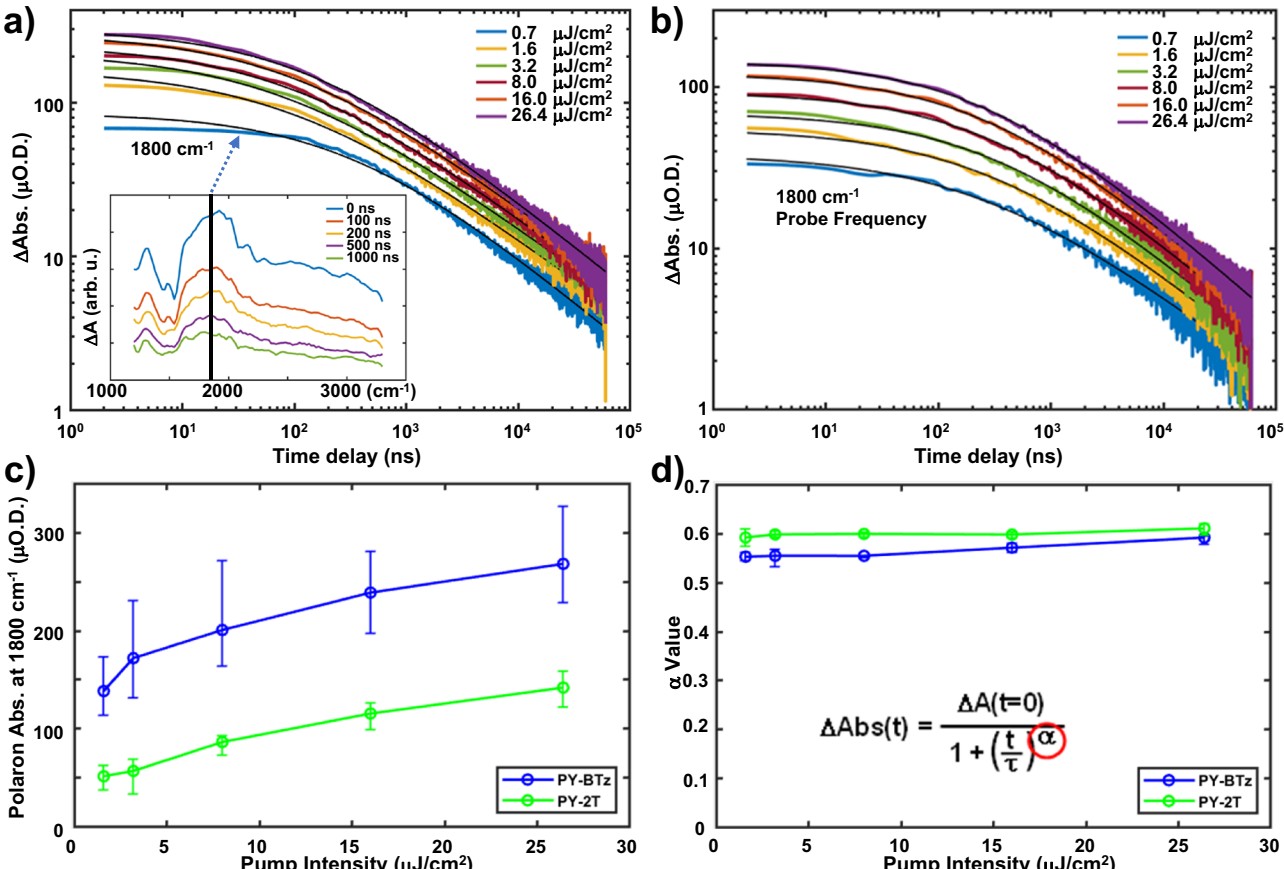

**Fig. 6 | Nanosecond transient absorption experiments in the IR.** Unnormalized transient absorption decay kinetics of polarons formed in (**a**) PY-BTz and (**b**) PY-2T based blend films following light excitation at 532 nm with different pump pulse intensities. The solid lines on top of the experimental data correspond to power law fit functions. Plot of (**c**) signal intensity and (**d**) $\alpha$ values determined from the kinetics at several pump intensities for the PY-BTz based blend (blue lines) versus PY-2T based blend (green lines).

PBDB-TF$_{0.25}$:PY-2T blend thin films. The scattering data were collected at various photon energies[68], and those taken at 284.0 eV were used for the data analysis because they provided maximum scattering contrast between the components. The domain sizes estimated by $2\pi/q_{peak}$ are 17 and 20 nm for PBDB-TF$_{0.25}$:PY-BTz and PBDB-TF$_{0.25}$:PY-2T blend films, respectively, consistent with AFM and other results. The slightly reduced phase separation and increased polymer intermixing for PBDB-TF$_{0.25}$:PY-BTz, as indicated by its lower R-SoXS scattering intensity compared to that of PBDB-TF$_{0.25}$:PY-2T blend films, agree well with higher $J_{SC}$, improved PL quenching efficiency, and higher photoinduced hole transfer yield results.

To further understand the effect of the molecular structure on the film morphology, we performed in situ UV–Vis absorption spectroscopy during the spin-coating of PY-BTz and PY-2T blend films (Supplementary Fig. 28) and their drying kinetics. The spectra were acquired over the wavelength range of interest, from 380 nm to 1000 nm, and with an integration time of 0.1 s. Since the donor polymer is unchanged, here we focus our discussions on the acceptor polymers. The in situ spectral changes show the transition from the solution state to the thin film. We first observed a continuous increase in the intensity ratio between the intrachain ($A_{0-1}$) and interchain ($A_{0-0}$) vibronic transitions, which is an indication of increased aggregation of the polymer. The evolution of the absorption spectra can be divided into three stages. In the earliest stage, a dramatic reduction in the absorption intensity of the solution peak can be observed due to large loss of the initial volume of the solution caused by ejection during spining[69]. The following stage shows a steady absorption of the solution peak, where the solvent evaporation takes place without

significant change in the solution state of the polymers. In the last stage, the solid thin film is formed, as indicated by the dramatic increase in the polymer absorption features associated with aggregation and subsequently reaches a plateau. Since the spin-coating is performed under the same experimental conditions, any differences in the kinetics from the two solutions can provide information about the drying and polymer aggregation properties., This is done by monitoring the absolute absorbance and intensity ratio between the intrachain (0–1) and interchain (0–0) vibronic transitions. It takes $21.0 \pm 0.3$ s for the PY-BTz blend solution to form a dry film, but only $18.1 \pm 0.4$ s for the PY-2T solution (Supplementary Fig. 28c). The longer drying kinetics in PY-BTz system may partially explain its stronger crystallinity and order. Moreover, Supplementary Fig. 28d shows that the PY-BTz blend film has a significantly higher $A_{0-0}/A_{0-1}$ value than that of the PY-2T blend film throughout the spin-coating process. This observation corroborates the enhanced aggregation of PY-BTz blend. The increase in the $A_{0-0}/A_{0-1}$ ratio was further accompanied by a red shift of the $A_{0-0}$ peak (Supplementary Fig. 28e), implying a longer effective conjugation length is achieved. According to these results, we conclude that the nature of the π-linker has indeed a key role in the solution processed film formation, thus the PY-BTz chains can be more stretched to adopt extended conformations and allow a tight assembly into ordered structures in the donor polymer matrix upon film formation.

### Energy-loss analysis
Lastly, we performed a series of photophysical measurements to investigate energetic disorder within all-PSCs (Fig. 8)[70,71]. The total

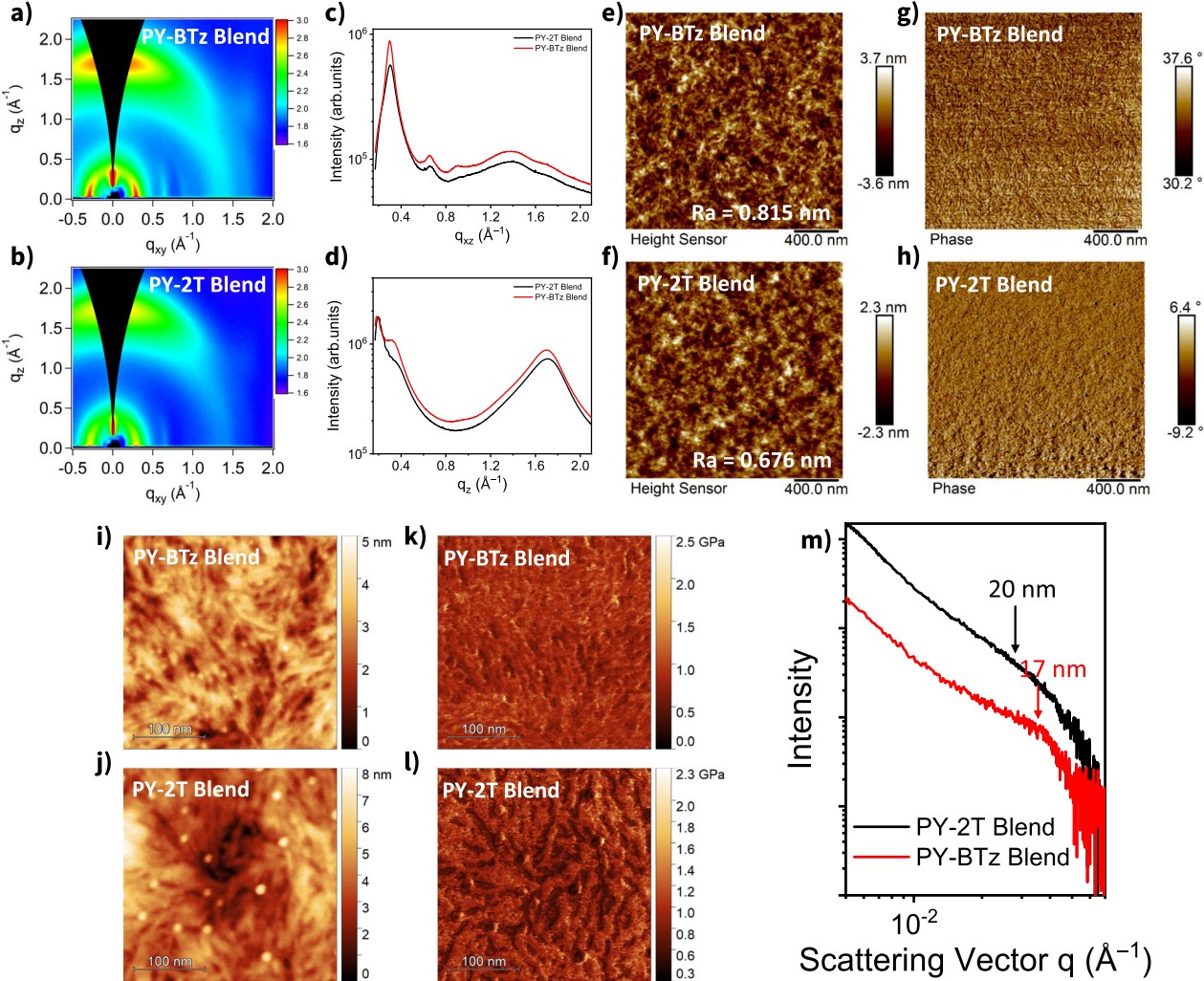

**Fig. 7 | Film morphology of blend films.** 2D-GIWAXS patterns of (**a**) PY-BTz and (**b**) PY-2T based blend films. **c** In-plane and (**d**) out-of-plane line cuts of for PY-BTz based blend film (red line) respect PY-2T based one (black line). AFM height images (2 × 2 μm) of (**e**) PY-BTz and (**f**) PY-2T based blend films obtained in tapping-mode. Corresponding phase images are shown in (**g**) and (**h**). **i–l** AFM height images (300 × 300 nm) and corresponding DMT (Derjaguin, Muller, Toporov) modulus image of blended thin film measured in peak force quantitative nanomechanical (QNM) mode. **m** R-SoXS profiles for PY-BTz based blend film (red line) respect PY-2T based one (black line).

energy losses ($\Delta E_{loss}$) were calculated to be 0.54 and 0.56 eV for PBDB-TF$_{0.25}$:PY-BTz and PBDB-TF$_{0.25}$:PY-2T, respectively. To further probe into more details, the three parts of $\Delta E_{loss}$ were determined according to detailed balance theory (Fig. 8c)[72,73]. $\Delta E_1$ arises from the radiative recombination loss from $E_g$ (band gap) and is unavoidable for solar cells. This value is found to be nearly comparable for the same polymer acceptor. $\Delta E_2$ results from the radiative recombination loss. It is influenced by the absorption below $E_g$, and the PY-BTz device exhibits a low value of 0.03 eV compared with the PY-2T devices (0.05 eV). The last part, $\Delta E_3$, is associated with nonradiative recombination only[70,71]. The smaller $\Delta E_2$, $\Delta E_3$, and $\Delta E_{loss}$ of PY-BTz based all-PSCs should be, therefore, attributed to the more regular molecular packing, reduced energetic and morphological disorder, and suppressed nonradiative recombination losses. Additionally, through dual fitting of electroluminescence (EL) and Fourier-transform photocurrent spectroscopy-external quantum efficiency (FTPS-EQE) (Fig. 8a, b), the charge-transfer state energy ($E_{CT}$) of the two all-PSCs was obtained. PBDB-TF$_{0.25}$:PY-BTz showed a relatively smaller $E_{CT}$ of 1.41 eV, in comparison to that of PBDB-TF$_{0.25}$:PY-2T (1.44 eV). $E_{CT}$ is related to the electron-cloud delocalization and electronic polarization effects that originate reduced nonradiative recombination loss[74,75]. Moreover, the smaller Urbach energy ($E_U$)[76], estimated from the tail region of either EQE or photothermal deflection spectroscopy (PDS), also indicates smaller energetic disorder of PY-BTz ($E_U^{EQE}$ = 20.6 meV or $E_U^{PDS}$ = 26.3 meV) based devices when compared to PY-2T devices ($E_U^{EQE}$ = 21.8 meV or $E_U^{PDS}$ = 27.3 meV) (Supplementary Fig. 29). The change of $E_U$ is correlated with a change in defect density of the film[77]. By combining these results with the molecular conformations and aggregation properties discussed earlier, it is reasonable to conclude that the smaller $E_{loss}$, $E_{CT}$, and $E_U$ are in fact indicative of higher degree of order of PY-BTz based systems, which suppresses molecular rotations and nonradiative recombination[78–80].

## Discussion

We have designed and synthesized a new polymer acceptor, PY-BTz, which incorporates a bithiazole linker instead of the often-used thiophene-based linker unit. The bithiazole linker minimizes torsional angles and thus increases backbone rigidity by the inclusion of non-covalent conformation lock effect and by removing C−H⋯H−C repulsions between neighboring arene units. A series of computational, X-ray scattering, and spectroscopic characterizations revealed that stronger conjugation and more planar polymer conformation are achieved in PY-BTz when compared to PY-2T, leading to an enhanced electron mobility (up to 5.4 ± 0.9 × 10⁻⁴ cm² V⁻¹ s⁻¹) and reduced

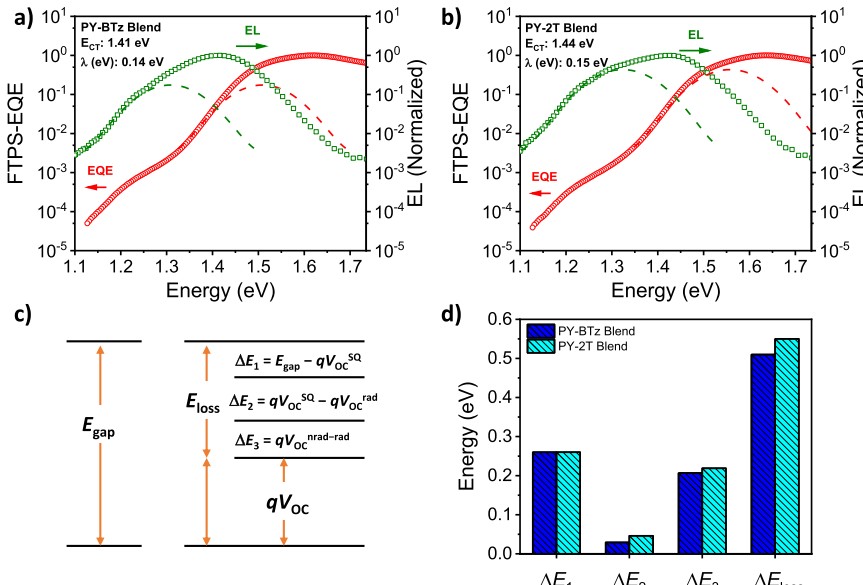

**Fig. 8 | Energy-loss analysis.** Semilogarithmic plots of normalized EQE calculated by Fourier transform photocurrent spectroscopy (FTPS-EQE) and electroluminescence (EL) curves of (**a**) PY-BTz and (**b**) PY-2T based all-PSCs. **c** Schematic diagram for energy losses ($E_{loss}$) of OSCs according to detailed balance. **d** $E_{loss}$ and its detailed three parts of $\Delta E_1$, $\Delta E_2$, and $\Delta E_3$ values.

conformational disorder (as demonstrated by the increased resistance toward thermally induced UV−Vis absorption spectra change, a red-shifted electronic absorption on-set, a closer π–π stacking distance, and reduced Urbach energy). When blended with a PBDB-TF$_{0.25}$ donor, combined morphological and device experiments showed that the PBDB-TF$_{0.25}$:PY-BTz system had a denser interchain packing and suitable phase separation features that leads to better exciton dissociation, faster charge transport, and reduced charge recombination losses. The in situ UV−vis measurements during film coating reveal that the nature of the π-linker has a vital role, where the PY-BTz polymer are more stretched to adopt extended planarized conformation that further tightly assemble into ordered structures in the donor polymer matrix during thin film formation. Photophysical investigations reveal that the PBDB-T$_{0.25}$:PY-BTz device possesses relatively smaller energy loss and Urbach energy due to the bithiazole-assisted coplanarization that reduces energetic disorder. Overall, the PBDB-TF$_{0.25}$:PY-BTz based all-PSCs delivers an enhanced efficiency of 16.0%, outperforming PBDB-TF$_{0.25}$:PY-2T (9.8%). Additionally, PBDB-TF$_{0.25}$:PY-BTz devices also achieve a record efficiency of over 14% in the solution sheared solar cells under ambient conditions, demonstrating the potential for large-scale industrial printing techniques. Our work illustrates how tuning π-linkage units can lead to more planarity and conformational rigidity via non-covalent interactions, which is an effective way to suppress energetic disorder in polymer semiconductors and to improving the PCE of printable all-PSCs. Our new materials design principle, in combination with new state-of-the-art processing engineering methods involving solid additives and solvent-vapor annealing post-treatments[81], will likely further improve all-PSCs performance.

## Methods
### Materials and instrumentation
All solvents were obtained from commercial suppliers and used without further purification unless stated otherwise. Instrumentation details can be found in the Supplementary Methods. Nuclear magnetic resonance (NMR) spectra were acquired on a Bruker Avance 500 spectrometer, with working frequencies of 500 MHz ($^1$H) and at 125 MHz ($^{13}$C), respectively. Chemical shifts are shown in ppm relative to the reference peak corresponding to the residual non-deuterated solvents ($\delta CDCl_3 = 7.26$ ppm, $\delta CD_2Cl_2 = 5.32$ ppm). MALDI

measurements were conducted on a Bruker Microflex MALDI-TOF spectrometer. Polymer samples were prepared by dissolving the polymers and the MALDI matrix (trans-2-[3-(4-tert-Butylphenyl)-2-methyl-2-propenylidene]malononitrile (DCTB)) in chlorobenzene and spotted the solution on the MALDI plate. Thermogravimetric analysis (TGA) and differential scanning calorimetry (DSC) measurements were carried out under continuous N$_2$ flow on a TA Instrument Q100 and Mettler Toledon AG-TGA/SDTA851e model, respectively. Electrochemical experiments were performed with a CH Instruments 600E potentiostat. Photoelectron spectroscopy in air (PESA) measurements were collected on a Riken AC-2 photoelectron spectrometer with a power setting of 5 nW and a power number of 0.5. Absorption spectra were collected using an Agilent Cary 6000i UV/Vis/NIR. Photoluminescence (PL) spectroscopy was performed using HORIBA Fluorolog3 spectrofluorometer.

Ultrafast transient absorption setup with a tunable pump and white-light probe was utilized to measure the transient differential absorption spectra through the thin film samples. The laser-based excitation system consists of a regeneratively amplified Ti:sapphire oscillator (Coherent Libra), which provides 4 mJ pulse energies centered at 800 nm with a 1 kHz repetition rate and ≈50 fs pulse duration. The light output was then split through an optical wedge to produce the pump and probe beams. The pump light was focused onto the sample by spherical lens at near-normal incidence with a spot size FWHM of ≈300 μm. The probe light was focused onto a sapphire plate to generate a white-light continuum probe, which was collected and refocused onto the sample by a spherical mirror (spot size FWHM -150 μm). The transmitted white light probe was acquired and processed with a commercial absorption spectrometer (Helios, Ultrafast Systems LLC). Pulse-to-pulse intensity fluctuations of the white light were adjusted for by simultaneously measuring a reference continuum. For our measurements, the pump wavelength was kept at 800 nm with a pulse power of 100 nJ (or -80 μJ/cm²). The pump and probe beams were linearly cross-polarized and any scattered pump-light into the detection path was filtered by a linear polarizer. The time delay was tuned by delaying the pump pulse with a linear translation stage (minimum step size 16 fs). For the data analysis, the individual component kinetic traces were fit to biexponential decays via least squares fitting unless stated otherwise. Nanosecond mid-IR transient

absorption experiments were performed using an inspIRe transient absorption spectrometer from Magnitude Instruments (State College, PA). This setup utilized the second harmonic generated by a nanosecond Nd:YAG laser (532 nm) to excite the samples. An infrared glowbar was utilized to generate the infrared probe light. The signal was collected using a liquid $N_2$ cooled mercury cadmium telluride (MCT) photovoltaic detector.

## General procedures for the synthesis of PY-BTz and PY-2T

Y6-OD-2Br (60 mg), 4-4′-bis(octyloxy)-2,2′-bis(trimethylstannyl)-5,5′-bithiazole (24.0 mg) or 5,5′-bis(trimethylstannyl)-2,2′-bithiophene (16.9 mg), $Pd_2(dba)_3$ (2 mol% vs. Y6-OD-2Br) and P(o-tolyl)$_3$ (8 mol% vs. Y6-OD-2Br) were combined in dry chlorobenzene (3 ml) in a nitrogen glovebox. The reaction mixtures, protected from light, were stirred at 120 °C for 2 days before being cooled down to room temperature. The crude product was precipitated out in methanol and collected. Then, it was purified by Soxhlet extraction with hot methanol, acetone, and hexane under nitrogen for 12 h each. PY-BTz, $M_n = 15.5$ kDa, $Đ_M = 2.64$; PY-2T, $M_n = 15.7$ kDa, $Đ_M = 2.62$. The final product fraction was collected and dried under vacuum before any characterization. MALDI-TOF, FTIR, and NMR spectra of PY-BTz and PY-2T are shown in Supplementary Figs. 2–9, respectively.

## Procedures for the synthesis of PBDB-TF$_x$

different molar fractions of the fluorine-containing monomer units, 2,6-bis(trimethylstannyl)-4,8-bis(5-(2-ethylhexyl)-4fluorothiophen-2-yl)benzo[1,2-b:4,5-b′]dithiophene (0, 30.7, 61.3, 92.0, 122.7 mg, respectively) was used as the third monomer in the copolymerization reaction via Stille coupling polycondensation between 2,6-bis(trimethytin)-4,8-bis(5-(2-ethylhexyl)thiophen-2-yl)benzo[1,2-b:4,5-b′]dithiophene (118, 88.5, 59, 29.5, 0 mg, respectively) and 1,3-bis(5-bromothiophen-2-yl)-5,7-bis(2-ethylhexyl)benzo[1,2-c:4,5-c′]dithiophene-4,8-dione (100 mg), affording PBDB-TF$_x$ (Supplementary Fig. 1, x = 0, 0.25, 0.5, 0.75, 1, respectively). $Pd_2(dba)_3$ (2 mol% vs. Y6-OD-2Br) and P(o-tolyl)$_3$ (8 mol% vs. Y6-OD-2Br) were added to the chlorobenzene solution (15 ml) containing the monomers, and the rection mixtures were stirred 120 °C for 48 h. These polymers share the same solubilizing side-chains to ensure good solubility, while the backbone fluorine substitution is expected to fine-tune the frontier orbital energy levels and the polymer:polymer blend thin films morphology.

## General computational settings

All DFT calculations have been performed using the ORCA quantum chemistry program package (5.0.3 version). All geometry optimizations of monomer, dimer, and trimer PY-2T and PY-BTz structures have been carried out in vacuo using the B3LYP exchange-correlation functional, the double-zeta quality def2-SVP basis set and the D3 atom-pairwise dispersion correction with the Becke-Johnson damping scheme (D3BJ). In all cases, structures have been simplified by replacing the alkyl side chains with methyl groups.

All properties (i.e., absolute energies, molecular electrostatic potential, and atomic charges) have been evaluated by carrying out a single point calculation on optimized structures at the same level of theory but with a triple-zeta quality def2-TZVPP basis set. Relaxed potential energy surface scans have been carried out with the same protocol (i.e., scanning at the def2-SVP level and refining the relative energies with the larger def2-TZVPP basis set). Atomic charges on the bithiophene/bithiazole units of PY-2T/PY-BTz monomers have been computed from the molecular electrostatic potential by using the Charges from Electrostatic Potentials using a Grid-based (CHELPG) method as implemented in ORCA.

## Morphology characterizations

Detailed description of GIWAXS, RSoXS, and AFM methods can be found in the Supporting Information. Briefly, GIWAXS experiments were collected at beamline 11-3 of the Stanford Synchrotron Radiation Lightsource (SSRL) using 2D area detector in a helium chamber to minimize air-scattering background. RSoXS measurements were carried out at Advanced Light Source (ALS) beamline 11.0.1.2 in transmission geometry under vacuum. Data analysis was also performed using the Nika package supported in the Igor Pro environment. The AFM topographical images were acquired using a Bruker Dimension Icon atomic force microscope in tapping mode and NSC15/Al-BS (MikroMasch, Tallinn, Estonia) cantilever (typical resonant frequency of 325 kHz and force constant of 40 N m$^{-1}$). The root-mean-square (RMS) surface roughness and phase differences were calculated over a $2 \times 2 \, \mu m^2$ area images acquired with $512 \times 512$ pixels resolution and 0.8 Hz scan rate. AFM-QNM images were collected using a Bruker Icon Dimension with NanoScope V electronics and a HQ:NSC19/Al BS probe (from MikroMasch with a nominal spring constant of 0.5 N/m, resonance frequency of 65 kHz, and a tip radius of 8 nm) which were calibrated right before the sample measurements. The calibration protocol involved the determination of the force constant (0.75 N/m) via thermal tuning, the deflection sensitivity (75.47 nm/V) against a sapphire reference sample, and the tip radius (3.16 nm) via tip qualification (NanoScope Analysis 3.0, Bruker) on a rough Ti reference sample. Nanomechanical AMF images were acquired in the Peak Force Quantitative Nanomechanics (PF-QNM) mode at a setpoint of ~800 pN, with a Peak Force frequency of 2 kHz, and amplitude of 150 nm. The image resolution was set to be $512 \times 512$ pixel (at a scan-rate of 0.5 Hz) for the 1 um images and $256 \times 256$ pixels (at 0.7 Hz) for the 300 nm images. The data were analyzed and plotted with Gwyddion SPM software.

## Solar cell fabrication and testing

Details on the device fabrication and characterization (solar cells, charge mobility, EQE, EL) are reported in the Supporting Methods. The ITO patterned ITO glass substrates (20 mm × 20 mm, with a sheet resistance of 13 Ω/square) were purchased from Xin Yan Technology Ltd. and used as received. Following the UV–ozone treatment of the substrates for 20 min, ZnO layer was fabricated by first depositing a diethylzinc precursor solution (15 wt% diethylzinc solution diluted 1:7 with tetrahydrofuran, both purchased from Sigma-Aldrich) onto the ITO surface via spin-coating at a speed of 5000 rpm for 30 s. The 30 nm thick ZnO film was produced after baking at 200 °C for 0.5 h in air. The polymer samples were dissolved in chlorobenzene (15 mg ml$^{-1}$ for donor and acceptors combined and 1:1 ratio by weight) and stirred at 80 °C for at least 8 h in a nitrogen glovebox. 1-chloronaphthalene (0.75% v/v) was added to the solution just before the film deposition as solvent additive, and the thickness of the spin-coated active layer film was around 110 nm (1000 rpm for 1 min in air or $N_2$, followed by drying in $N_2$ for 2 min at 150 °C temperature). A $MoO_3$ hole-transport layer (7.5 nm) followed by an Ag layer (100 nm) were thermally deposited at a pressure of ca. $8 \times 10^{-6}$ Torr using a vacuum evaporator. All solar cells were analyzed inside a nitrogen glove box under AM 1.5G illumination with an intensity of 100 mW cm$^{-2}$ (Newport Solar Simulator 94021A). The setup calibration was performed using a Newport certified silicon photodiode covered with a KG5 filter and with an active area of 6.63 mm$^2$, which is comparable to the device effective area of 4.0 mm$^2$. The $J–V$ curves were collected with a Keithley's Standard Series 2400 Source Measure Unit (SMU) Instrument. External quantum efficiency experiments were performed at short circuit using monochromated light from a tungsten lamp that was modulated by an optical chopper. The current from the samples was collected as a function of excitation wavelength and compared to the current obtained from a photodiode with an NIST traceable calibration photocurrent action spectrum.

For the charge mobility determination, single carrier devices were fabricated, and the dark $J–V$ characteristics measured and analyzed in the space charge limited current (SCLC) regime. Specifically, the structure of hole- and electron-only devices was Glass/ITO/

PEDOT:PSS/Active layer/MoO$_3$/Ag and Glass/ITO/ZnO/Active layer/LiF/Al, respectively. Mobilities were calculated by fitting the $J$–$V$ curves using the Mott-Gurney relationship. Highly sensitive FTPS-EQE measurements were carried out following the procedure outlined by Vandewal et al., using a Nicolet iS50R FT-IR connected to a SRS Model SR570 low-noise current preamplifier. For the electroluminescence experiments, current was injected using a Keithley 2400 SMU, and emitted photons were detected with the Ocean Insight QE Pro Spectrometer.

## Single crystal XRD experimental

Single crystals of C$_{25.5}$H$_{33}$N$_{1.5}$O$_{1.5}$S$_{1.5}$ [Ph-BTz, CCDC 2269848, Supplementary Data 1 and 2] were obtained by slow vapor diffusion of methanol in chlorobenzene solution. A suitable crystal was picked and mounted on a Bruker D8 Venture single-crystal X-ray diffractometer using Mo radiation. The sample was kept at 100.00 K throughout data collection. Using Olex2 software, the structure was solved with the SHELXS structure solution program using Direct Methods. The structure was refined with the SHELXL refinement package using Least Squares minimization.

## Crystal data

for C$_{25.5}$H$_{33}$N$_{1.5}$O$_{1.5}$S$_{1.5}$ ($M$ = 432.62 g/mol): triclinic, space group P-1 (no. 2), $a$ = 11.957(3) Å, $b$ = 12.529(4) Å, $c$ = 17.126(6) Å, $\alpha$ = 86.016(7)°, $\beta$ = 84.547(7)°, $\gamma$ = 63.378(9)°, $V$ = 2282.3(12) Å$^3$, $Z$ = 4, $T$ = 100.00 K, $\mu$(MoK$\alpha$) = 0.209 mm$^{-1}$, $D$calc = 1.259 g/cm$^3$, 164,783 reflections measured (3.638° ≤ 2Θ ≤ 63.7°), 15,644 unique ($R_{int}$ = 0.1144, $R_{sigma}$ = 0.0582) which were used in all calculations. The final $R_1$ was 0.0535 ($I$ > 2$\sigma$($I$)) and $wR_2$ was 0.1902 (all data).

## Reporting summary

Further information on research design is available in the Nature Portfolio Reporting Summary linked to this article.

# Data availability

The experiment data that support the findings of this study are available from the corresponding author upon request. Crystallographic data for the structures reported in this Article have been deposited at the Cambridge Crystallographic Data Centre, under deposition numbers CCDC 2269848. Copies of the data can be obtained free of charge via https://www.ccdc.cam.ac.uk/structures/. Source data are provided with this paper.

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

## Acknowledgements

The authors acknowledge financial support from the Office of Naval Research Multidisciplinary University Research Initiatives (MURI) Program (Program manager P. Armistead, award N00014-19-1-2453). Use of the Stanford Synchrotron Radiation Lightsource, SLAC National Accelerator Laboratory, is supported by the U.S. Department of Energy, Office of Science, Office of Basic Energy Sciences under Contract No. DE-AC02-76SF00515. Part of this work was performed at nano@stanford, supported by the National Science Foundation under award ECCS-2026822. We thank Jerika Chiong and Yu Zheng for performing the high-temperature SEC experiments. L.M. gratefully acknowledges funding through the Walter Benjamin Fellowship Programme by the Deutsche Forschungsgemeinschaft (DFG 456522816).

## Author contributions

The project was conceived by Y.W. and Z.B. Y.W. conducted experiments involving synthesis, characterization, GIWAXS, AFM, and device fabrication. L.M. conducted QNM imaging. H.C. and S.Z. conducted experiments involving solution coating and device fabrication. Y.Y. and J.B.A. conducted the TRIR. D.S., G.G., V.J. and S.M. conducted the DFT calculation. C.C., G.L. and A.S. conducted the FTPS-EQE and EL experiment. E.D.G. and M.F.T. contributed to the interpretation of the results. All authors participated in the result analysis and contributed to the editing of the manuscript.

## Competing interests

J.B.A. owns equity in Magnitude Instruments, which has an interest in this project. His ownership in this company has been reviewed by the Pennsylvania State University's Individual Conflict of Interest Committee and is currently being managed by the University. The remaining authors declare no competing interests.
