## [Peer Review File · Nature Communications]

Tuning Polymer-Backbone Coplanarity and Conformational Order to Achieve High-Performance Printed All-Polymer Solar CellsREVIEWER COMMENTS

Reviewer #1 (Remarks to the Author):

Yilei Wu et al. designed and synthesized two new polymer acceptor namely PY-2T and PY-BTz. by linking Y6-analogue unit with bithiazole unit or thiophene unit. In bithiazole case, the polymer demonstrated more coplanar and ordered molecular conformation thanks to the non-covalent backbone planarization and reduced steric encumbrances. Based on PY-BTz, the devices show a decent efficiency of 16.4% due to tighter interchain π -stacking, leading to higher charge mobilities, suppressed charge recombination, and reduced energetic disorder. Although the efficiencies still lags behind those from the state-of-the-art ones especially the recent 19%, the design rule is very helpful to this field. I can recommend its publication after my concerns are adequately addressed.

(1) A systematical and solid correlation should be given between the more coplanar structure and the final device performance. A more solid connection among more coplanar structure, tighter interchain π -stacking, leading to higher charge mobilities, suppressed charge recombination, and reduced energetic disorder should be provided.

(2) As we know, the best-performing all polymer solar cell devices are based on PY-IT, showing 19% efficiency. Is the design rule including more coplanar and rigidity applied to PY-IT. Systematical study and discussion on the design rule among the three materials should be provided because this strongly correlates with the novelty of this manuscript.

(3) A more detailed morphology should be given, which can help the authors to further understand the superiority of PY-BTz than PY-2T. Giwax can only give simple crystallinity information.

(4) In Scheme 1, the authors stated "less steric hindrance" in PY-BTz. Usually, "less steric hindrance" indicates the rotation between units becomes easier, thus decreasing the planarity. More discussion should be given regarding this statement including a good comparison between the two polymers.

(5) The average molecular weight is only 13 kDa, which means the so called "polymers" are "oligomers" with about 4 to 5 repeating units.

(6) A more coplanar structure is computed in vacuum state. A more coplanar structure might be not the same case in the film state. In OPV blend, it contains the complicated interaction from D-A, A-A, D-D et al. How do the authors think about it.

Reviewer #2 (Remarks to the Author):

See Attachment

In this work, the authors synthesized a new polymer acceptor, PY-BTz, and realized an efficiency of 16.4% in all-polymer solar cells. The authors investigate the underlying charge recombination mechanism using various tools, including femtosecond transient absorption spectroscopy, Fourier transform photocurrent spectroscopy-EQE, and photoluminescence quenching efficiency. However, I see the work falling short in various interpretations. Therefore, this manuscript can be published in Nature Communications after following major revisions.

1. What is the reproducibility of PY-BTz synthesis? Does the device based on this polymer have the batch issue?
2. New polymers named PBDB-TFx were designed and synthesized. The authors should provide the synthetic information and device data of PBDB-TFx.
3. What is the difference between PBDB-TF and PBDB-TFx, which is not explained in the manuscript.
4. The device based on PY-BTz was given to have a PCE of only 16.4%, which is clearly lower than the highest performance for all-PSCs (19.06%, All-polymer organic solar cells with nano-to-micron hierarchical morphology and large light receiving angle, Nat. Commun., 2023,14, 4148.). The authors should discuss the limitation of this work and how to improve the all-PSCs.
5. Most of the high PCE values are not reproducible due to the lack of experimental details. The authors should include experimental details.
6. The authors only provided thermal stability of blend films. Is the stability higher than other reported all-PSCs? Photostability of the device (AM 1.5 G) is suggested to be provided.
7. It is not possible to get the exciton dissociation probabilities from curves in Figure S20. Photocurrents are not only determined by a field dependent charge generation but also by charge transport. To accurately measure charge generation yields, one would need time-delayed collection field or other techniques. Authors cannot rely on just a voltage-dependent measurement of the photocurrent.
8. The authors claim that PY-BTz reduces charge recombination in the title and the introduction. However, the FF of the device is only 0.67 ± 0.02 , which is a low value

in PV cells.

9. The authors emphasize scalable processing devices in this work. I do not find any more data except for the PV parameter.

10. Please provide the integral values of the EQE curves.

11. The $E_{U^{EQE}}$ of the device looks weird (lower than nKT). Please double-check the data fitting.

Reviewer #3 (Remarks to the Author):

The manuscript by Wu and coworkers discussed the effect of backbone coplanarity and conformational order in all polymer solar cells. This is an important topic, however only one pair of comparison results are given, and the planarity and conformation is assessed by calculations. These are not enough evidence to made such a broad claim, and the conclusions are not helpful in new material design. The literature survey is also dated, and the presented good case is only moderate performance in all polymer solar cells.

one more important issue is that how the backbone planarity effect is translated to thin film morphology is not well explained. Also no detailed discussions related with device performance. I would not suggest this is a topnotch research, and much more effort is needed.

Response to the referee reports regarding our manuscript entitled “*Tuning Polymer-Acceptor Backbone Coplanarity and Conformational Order to Achieve High-Performance Printed All-Polymer Solar Cells with Reduced Charge Recombination*”

Reviewer 1

Comments: *Yilei Wu et al. designed and synthesized two new polymer acceptor namely PY-2T and PY-BTz, by linking Y6-analogue unit with bithiazole unit or thiophene unit. In bithiazole case, the polymer demonstrated more coplanar and ordered molecular conformation thanks to the non-covalent backbone planarization and reduced steric encumbrances. Based on PY-BTz, the devices show a decent efficiency of 16.4% due to tighter interchain π -stacking, leading to higher charge mobilities, suppressed charge recombination, and reduced energetic disorder. Although the efficiencies still lags behind those from the state-of-the-art ones especially the recent 19%, the design rule is very helpful to this field. I can recommend its publication after my concerns are adequately addressed.*

Response: We thank Referee 1 for their positive review and for pointing out that “*Although the efficiencies still lags behind those from the state-of-the-art ones especially the recent 19%, the design rule is very helpful to this field*”. Below, we have outlined in detail the changes we made to our manuscript in order to address reviewer suggestions.

- 1) *A systematical and solid correlation should be given between the more coplanar structure and the final device performance. A more solid connection among more coplanar structure, tighter interchain π -stacking, leading to higher charge mobilities, suppressed charge recombination, and reduced energetic disorder should be provided?*

Response: We thank the reviewer for the suggestion. We have added the following statements in the conclusion section to strengthen such connections and correlations. “The bithiazole linker minimizes torsional angles and thus increases backbone rigidity by the introduction of non-covalent conformation lock effect and by eliminating repulsive C–H \cdots H–C interactions with neighboring arene units. A series of computational, X-ray scattering, and spectroscopic experiments indicated that stronger conjugation and more planar chain conformation are realized in PY-BTz when compared to PY-2T, leading to an increased electron mobility (up to $5.4 \pm 0.9 \times 10^{-4} \text{ cm}^2 \text{ V}^{-1} \text{ s}^{-1}$) and reduced conformational disorder (as demonstrated by the increased resistance toward thermally induced UV-Vis absorption spectra change, a red-shifted electronic absorption on-set, a closer π - π stacking distance, and reduced Urbach energy). When blended with a PBDB-TF_{0.25} donor, both morphological and device studies revealed that the PBDB-TF_{0.25}:PY-BTz blend formed a denser interchain packing and suitable phase segregation that leads to superior exciton dissociation, charge transport, and suppressed charge recombination. The in-situ UV-vis measurements during film coating reveal that the nature of the π -linker plays a vital role, where the PY-BTz chains are more stretched to adopt extended conformation and further tightly pack into ordered aggregates in the donor polymer matrix during film formation.”. We have also further expanded the critical role of the π -linker on the film morphology and film formation dynamics (see Response 3).

- 2) *As we know, the best-performing all polymer solar cell devices are based on PY-IT, showing 19% efficiency. Is the design rule including more coplanar and rigidity applied to PY-IT. Systematical*

study and discussion on the design rule among the three materials should be provided because this strongly correlates with the novelty of this manuscript.

Response: PY-IT does not follow the co-planar backbone design rule, which is the focus of our study.

- 3) *A more detailed morphology should be given, which can help the authors to further understand the superiority of PY-BTz than PY-2T. Giwax can only give simple crystallinity information?*

Response: We thank the reviewer for the suggestion. We have added more detailed morphology analysis as suggested and updated Figure 6 as follows:

Morphology characterization of Blend Films. To correlate the charge generation and transport properties with the morphology of the blends, GIWAXS, Atomic force microscopy (AFM), resonant soft X-ray scattering (R-SoXS), and transmission electron microscopy (HRTEM) were carried out. GIWAXS experiments were first conducted to investigate the microstructures and molecular packing behaviors of the blend films (**Figure 6a,b**).⁶² After blending with PBDB-TF_{0.25}, both PY-BTz and PY-2T based blends exhibit predominant face-on orientation with the π - π (010) stacking peaks located at ca. 1.67 \AA^{-1} in the out-of-plane direction (**Figure 6d**) and the lamellar (100) peaks at $q_z \approx 0.30 \text{ \AA}^{-1}$ in the in-plane direction (**Figure 6c**). This shows that the face-on orientation is dominant in the blends as for their neat films, which is beneficial for charge transport in diode configurations. However, PY-BTz based pristine and blend films both exhibit slightly larger crystal coherence lengths (CCL₁₀₀ and CCL₀₁₀) than the PY-2T systems, suggesting a relatively higher order of PY-BTz (**Figure S25** and **Table S3**), which is beneficial for charge transport.

AFM imaging of the active layer was performed to study the surface morphology. The PY-BTz-based blends exhibit a slightly larger root-mean-square roughness compared to PY-2T-based ones, which is beneficial for increasing the contact area with electrodes and therefore improves extraction (**Figure 6e-f**). More importantly, both the height and phase images (**Figure 6g-h**) show

Figure 6. Film morphology of blend films. 2D-GIWAXS patterns of a) PY-BTz and b) PY-2T based blend films. c) In-plane and d) out-of-plane line cuts of for PY-BTz based blend film (red line) respect PY-2T based one (black line). AFM height images ($2 \times 2 \mu\text{m}$) of e) PY-BTz and f) PY-2T based blend films obtained in tapping-mode. Corresponding phase images are shown in g) and h). i-l) AFM height images ($300 \times 300 \text{ nm}$) and corresponding DMT (Derjaguin, Muller, Toporov) modulus image of blended thin film measured in peak force quantitative nanomechanical (QNM) mode. m) R-SoXS profiles for PY-BTz based blend film (red line) respect PY-2T based one (black line).

that the PY-BTz-based blend films also possess slightly more well-defined and finer fibrillar morphology, which could contribute to better charge transport and fill factor. To further understand the nature of the detected domains, DMT (Derjaguin–Muller–Toporov) modulus⁶³ images were acquired via PeakForce Quantitative Nanomechanical property Mapping (QNM; **Figure 6k-l** and **Figure S26**). Since both the observed domains and the surroundings show similar DMT moduli (ca. 1 GPa), no major compositional differences are to be expected. Here, the AFM images clearly show a continuous interpenetrating network with a feature size in the range of 10–20nm—a morphology clearly favourable for exciton dissociation and charge transport. Remarkably, a small

reduction of the domain size is observed in PY-BTz-based blend films compared to PY-2T blends, giving rise to improved solar cell performances (**Table 2** and **Figure 3**).

To measure changes in the internal structure of blend we performed transmission R-SoXS measurements. This technique is known to enhance the contrast between two constituent polymers and provide information on the degree of nanoscale phase separation.^{64–68} **Figure 6m** shows R-SoXS profiles of PBDB-TF_{0.25}:PY-BTz and PBDB-TF_{0.25}:PY-2T blend films. The scattering data were taken at different photon energies,⁶⁹ and those acquired at 284.0 eV were used for the analysis as they provided maximum scattering contrast between donor and acceptor. The domain sizes estimated by $2\pi/q_{\text{peak}}$ are 17 and 20 nm for PBDB-TF_{0.25}:PY-BTz and PBDB-TF_{0.25}:PY-2T blend films, respectively, consistent with AFM and other results. The slightly reduced domain sizes and increased polymer intermixing for PBDB-TF_{0.25}:PY-BTz, as suggested by its reduced R-SoXS scattering intensity PBDB-TF_{0.25}:PY-2T blend films, agree well with higher J_{SC} , improved PL quenching efficiency, and higher photoinduced hole transfer yield results.

To better understand the effect of the molecular structure on the film morphology we performed in-situ UV-Vis absorption spectroscopy during the spin-coating of PY-BTz and PY-2T based blend films (**Figure S27**) to detail the difference in drying kinetics. The measurements were performed using an F20-EXR spectrometer (Filmetrics, Inc.) equipped with tungsten halogen and deuterium light sources over the wavelength range of interest, from 400 nm to 1000 nm. The measurements were performed with an integration time of 0.1 s. Since the donor polymer is the same, here we will focus our discussion on the acceptor polymer. The spectra show the phase transition from the dissolved state to the solid state, starting with a continuous increase in the intensity ratio between the intrachain (A_{0-1}) and interchain (A_{0-0}) vibronic transitions, indicating an increased aggregation of the polymer. The evolution of the absorption spectra follows three

distinct stages. In the first stage, we observe a steady and dramatic reduction in the absorption intensity of the solution peak, as a result of loss of the initial volume of the solution via ejection.⁷⁰ The second stage is characterized by a steady absorption of the solution peak, indicating the solvent evaporation process is underway without inducing significant change in the solution state of the polymers. The third stage corresponds to the solid-state thin film formation process, whereby the polymer absorption peaks associated with aggregation increase dramatically and subsequently reach a plateau. As the spin-coating progresses under identical conditions, we observe differences in the kinetics of the drying and polymer aggregation from the two solutions, by monitoring the absolute absorbance and intensity ratio between the intrachain (0–1) and interchain (0–0) vibronic transitions. It takes 21.0 ± 0.3 seconds for the PY-BTz blend solution to form a dry film and only 18.1 ± 0.4 seconds for the PY-2T solution (**Figure S27c**). The longer drying kinetics in PY-BTz system may partially explain its stronger crystallinity/order. Moreover, **Figure S27d** shows that the PY-BTz blend film has a significantly higher A_{0-0}/A_{0-1} value than that of the PY-2T blend film throughout the spin-coating process. This indicates enhanced aggregation of PY-BTz blend. This increase in the A_{0-0}/A_{0-1} value was also accompanied by a red shift of the A_{0-0} peak (**Figure S27e**), indicating a longer effective conjugation length. Based on these results, we conclude that the nature of the π -linker plays a vital role in the solution processed blend films, where the PY-BTz chains are more stretched to adopt extended conformation and further tightly pack into ordered aggregates in the donor polymer matrix during film formation.

Figure S27. In-situ UV-vis absorption spectra during spin-coating of all-PSCs based on PBDB-TF_{0.25}:PY-BTz and PBDB-TF_{0.25}:PY-2T. Morphology evolution kinetics by tracking changes in c) absorbance of the peaks, d) ratio of 0-0 and 0-1 peaks of the acceptor polymer, and e) wavelength of 0-0 peaks.

- 4) *In Scheme 1, the authors stated “less steric hindrance” in PY-BTz. Usually, “less steric hindrance” indicates the rotation between units becomes easier, thus decreasing the planarity. More discussion should be given regarding this statement including a good comparison between the two polymers.*

Response: We thank the reviewer for making this point. We have changed the Scheme 1 accordingly and added following sentence in the scheme caption: “Compared with PY-2T, PY-BTz is designed to have a more co-planar and rigid polymer backbone thanks to: 1) the S(thiazolyl)···O(alkoxy) attraction promoting a non-covalent conformational lock effect and 2) the replacement of (thiophene)C–H with (thiazole)N eliminating repulsive C–H···H–C interactions with neighboring arene units”.

- 5) *The average molecular weight is only 13 kDa, which means the so called “polymers” are “oligomers” with about 4 to 5 repeating units.*

Response: We thank the reviewer for making this observation. We agree with the reviewer that the “polymer” is essentially with 4 to 5 repeating units since each monomer is rather large containing 9 aromatic units. Therefore, the total number of aromatic units (or length of the “polymer”) is comparable to other conjugated polymers with simple monomer structures.

- 6) *A more coplanar structure is computed in vacuum state. A more coplanar structure might be not the same case in the film state. In OPV blend, it contains the complicated interaction from D-A, A-A, D-D et al. How do the authors think about it.*

Response: We agree with the reviewer that the structure computed in vacuum might not always reflect the solid-state one. However, the computed torsional potential energy surfaces (Figure 1e-f) indicate that the presence of the non-covalent conformation lock in PY-BTz, combined with removal of repulsive (C-H)-(C-H) interactions, hinders bond rotation, suggesting that the molecular design proposed favors an overall more coplanar backbone for PY-BTz. Notably, these results are fully consistent with experimental XRD, FTIR and UV-Vis measurements.

Reviewer 2

Comments: *In this work, the authors synthesized a new polymer acceptor, PY-BTz, and realized an efficiency of 16.4% in all-polymer solar cells. The authors investigate the underlying charge recombination mechanism using various tools, including femtosecond transient absorption spectroscopy, Fourier transform photocurrent spectroscopy-EQE, and photoluminescence quenching efficiency. However, I see the work falling short in various interpretations. Therefore, this manuscript can be published in Nature Communications after following major revisions.*

Response: We thank this Reviewer for the positive comments. We have addressed the points that Reviewer 2 has indicated below.

- 1) *What is the reproducibility of PY-BTz synthesis? Does the device based on this polymer have the batch issue?*

Response: We have not experienced any reproducibility issues on the synthesis as it follows very established and robust methodologies. Moreover, no significant batch-to-batch variations in device performance have been observed. We synthesized 2 batches and the obtained average results and standard deviations are shown in table below:

Table S1. Batch-to-batch variation analysis of photovoltaic parameters of the All-PSC devices based on PBDB-TF_{0.25}:PY-BTz under the illumination of AM 1.5 G, 100 mW cm⁻².

Acceptor	Batch no.	Scale (mg)	M_n (kDa)	\bar{D}_M	J_{sc} (mA/cm ²)	V_{oc} (V)	Fill Factor	PCE _{ave} (%)
PY-BTz	1	60	15.5	2.64	26.5 ± 0.3	0.921 ± 0.007	0.646 ± 0.01	15.7 ± 0.2
PY-BTz	2	60	14.2	2.59	26.1 ± 0.5	0.913 ± 0.002	0.673 ± 0.01	16.0 ± 0.2

^aThe average values with standard deviations were obtained 15 (batch no 1) and 28 samples (batch no 2). Device structure of ITO/ZnO/PFNBr/Active-layer/MoO₃/Ag. Device area: 4 mm².

- 2) *New polymers named PBDB-TFx were designed and synthesized. The authors should provide the synthetic information and device data of PBDB-TFx?*

Response: We have now included the requested information regarding PBDB-TFx in the SI.

3) *What is the difference between PBDB-TF and PBDB-TFx, which is not explained in the manuscript.*

Response: PBDB-TF, known also as PM6 in literature, is equivalent to our PBDB-TF₁. We added this clarification in the paper.

4) *The device based on PY-BTz was given to have a PCE of only 16.4%, which is clearly lower than the highest performance for all-PSCs (19.06%, All-polymer organic solar cells with nano-to-micron hierarchical morphology and large light receiving angle, Nat. Commun., 2023,14, 4148.). The authors should discuss the limitation of this work and how to improve the all-PSCs.*

Response: We thank the reviewer for raising this comment. In the manuscript mentioned above (Nat. Commun., 2023,14, 4148.), Zheng and coworkers have developed a new sequential processing method, involving an innovative combination of solid additive of 1,4-diiodobenzene (DIB), thermal annealing and solvent vapor annealing post-treatments, to induce nano-to-micron sized hierarchical morphology. The authors demonstrated an impressive improvement of PCE for hierarchical morphology (18.32% ($V_{OC} = 0.944$ V, $J_{SC} = 25.92$ mA cm⁻², FF = 74.88%) compared to the conventional devices (15.17%, $V_{OC} = 0.933$ V, $J_{SC} = 24.12$ mA cm⁻², FF = 67.42%). Moreover, unprecedented high efficiency of 19.06% is obtained with a V_{OC} of 0.945 V, a J_{SC} of 26.37 mA cm⁻², and an FF of 76.48%, when the 2PACz was introduced as hole transporting layer (HTL). These more sophisticated processing optimization can potentially be applied to our new materials to further improve the device's performance in follow-up works. In this regard, we have added the following sentence in the conclusion section of the manuscript: "Our materials design principle, in combination with state-of-the-art processing engineering methods involving solid additives and solvent-vapor annealing post-treatments,^[Ref 81: Nat. Commun., 2023,14, 4148] and further device interface engineering, will likely give further improved all-PSCs performance."

5) *Most of the high PCE values are not reproducible due to the lack of experimental details. The authors should include experimental details.*

Response: Detailed "Solar Cell Fabrication and Testing" protocols are shown in section A of the Supplementary Information, along with other experimental details and error analysis.

6) *The authors only provided thermal stability of blend films. Is the stability higher than other reported all-PSCs? Photostability of the device (AM 1.5 G) is suggested to be provided.*

Response: Thermal stability is among the highest reported all-PSCs. See Table S3:

Table S2. Reported All-PSCs with high thermal stability.

Entry no.	Donor	Acceptor	Morphology	Initial PCE (%)	T (°C)	Time (h)	PCE Retention (%)	Reference
1	PBDT-TAZ	NOE10	Binary BHJ	8.1	65	300	97	J. Am. Chem. Soc. 2018, 140 , 8934-8943
2	PM6	PF1-TS4	Binary BHJ	8.63	80	180	70	Angew. Chem. Int. Ed. 2020, 59 , 19835-19840

3	PBDBT-BV20	N2200-TV10	Binary BHJ (crosslinked)	9.24	80	90	91	ACS Appl. Mater. Interfaces 2021, 13 , 16754-16765
4	PM6	PY2F-T:PYT	Ternary BHJ	17.2	65	300	83.5	Joule 2021, 5 , 1548-1565
5	PBDB-T	PTCloY	Binary BHJ	12.74	80	160	87	Sci China Chem 2022, 65 , 182-189
6	PBDB-T	PYT-H	Binary BHJ	14.29	80	408	96.96	J. Mater. Chem. C 2022, 10 , 1850-1861
7	PM6:PM6TPO	PY-IT	Ternary BHJ	17.0	80	96	85	Adv. Sci. 2022, 9 , 2204030

Photostability test (Figure S20) show > 90% PCE retention after over 1800 min continuous irradiation (100 mWcm^{-2}) in a nitrogen glove box, which is consistent with reported value for similar all-PSCs.

Figure S20. Normalized short-circuit current density, open-circuit voltage, fill factor, and PCE as a function of annealing time for blade-coated all-PSCs based on PBDB-TF_{0.25}:**PY-BTz** under continuous illumination (100 mWcm^{-2}) in a nitrogen glove box. The averages are from 5 individual devices.

- 7) *It is not possible to get the exciton dissociation probabilities from curves in Figure S20. Photocurrents are not only determined by a field dependent charge generation but also by charge transport. To accurately measure charge generation yields, one would need time-delayed collection field or other techniques. Authors cannot rely on just a voltage-dependent measurement of the photocurrent.*

Response: We estimated the exciton dissociation probabilities (P_{diss}) of the devices as the ratio of the photocurrent density (J_{ph}) to the saturation photocurrent density (J_{sat}) following the analytical approach reported by Wu et al. (*ACS Nano*, **2011**, *5*, 959–967) and many other (for example: *adv. Mater.* **2022**, *34*, 2108749; *adv. Mater.* **2017**, *29*, 1605115; *Nat Photonics* **2014**, *8*, 716-722; etc.), using the curves J_{ph} versus effective voltage (V_{eff}) (Figure S20).

“ J_{ph} is defined as $J_{ph} = J_L - J_D$, where J_L and J_D are the current densities of the devices under illumination and in the dark, respectively. V_{eff} is defined as $V_{eff} = V_0 - V_{bias}$, where V_0 is the voltage when J_{ph} is zero and V_{bias} is the applied voltage bias. Therefore, V_{eff} determines the electric field in the donor/ acceptor blend films and thus affects the exciton dissociation and charge transport behavior. The charge carriers move quickly to the corresponding electrodes and J_{ph} reaches saturation (J_{sat}) at a high V_{eff} ($V_{eff} \geq 2$ V), suggesting that all photogenerated carriers are extracted and collected with minimal recombination. Thus, the exciton dissociation probability of the PSCs can be calculated by J_{ph}/J_{sat} .”

We agree with the reviewer that this approach has its limitations. However, it should provide good relative values. Importantly, the higher P_{diss} for PY-BTz based All-PSC respect that of the PY-2T based one estimated using this method agrees well with the transient absorption results and PL quenching measurements. The determination of absolute exciton dissociation probabilities using transient electrical measurements is beyond the scope of this work.

- 8) *The authors claim that PY-BTz reduces charge recombination in the title and the introduction. However, the FF of the device is only 0.67 ± 0.02 , which is a low value in PV cells.*

Response: The 0.67 FF value for PY-BTz system is relatively good for all-PSCs, and significantly higher than PY-2T system (0.56). As a reference, the FF for recent record 19% PCE all-PSC (ref82) is 0.76. However, we believe further morphology and device interlayer optimization could further improve FF. We are currently working on these challenges.

- 9) *The authors emphasize scalable processing devices in this work. I do not find any more data except for the PV parameter.*

Response: In the paper, we reported PV cells fabricated in air using solution-shearing coating method, which mimics scalable processing conditions. In addition to PV parameters, we have shown that solution-shearing processed devices have improved thermal stability compared to the spin-coated ones. We speculate that this improvement in thermal stability might be due to relatively slower film drying kinetics of blade-coating method compared to spin-coating, resulting in fewer kinetic traps or instabilities in the blend film morphology. Further studies are currently underway to corroborate this hypothesis.

- 10) *Please provide the integral values of the EQE curves.*

Response: We thank the reviewer for the suggestion. We have now included these values (24.9 mA cm^{-2} for PBDB-TF_{0.25}:PY-BTz and 18.8 mA cm^{-2} for PBDB-TF_{0.25}:PY-2T) in the main text.

- 11) *The EUEQE of the device looks weird (lower than nKT). Please double-check the data fitting.*

Response: We have double-checked the data fitting and have not noticed any errors. While the actual values can be slightly dependent on the fitting method, the energy differences, however, are more reliable as long as the fitting procedure is consistent.

Reviewer 3

Comments: *The manuscript by Wu and coworkers discussed the effect of backbone coplanarity and conformational order in all polymer solar cells. This is an important topic, however only one pair of comparison results are given, and the planarity and conformation is assessed by calculations. These are not enough evidence to made such a broad claim, and the conclusions are not helpful in new material design. The literature survey is also dated, and the presented good case is only moderate performance in all polymer solar cells.*

Response: We thank the reviewer for the assessment. We have revised the manuscript according to all the reviewers' comments and we believe it is now significantly improved. We emphasize that this paper reports novel molecular design concepts. Combined with processing improvements and device engineering, it is likely to give further improvement of OPVs. We have also added more recent high-performing references as suggested.

Comments: *one more important issue is that how the backbone planarity effect is translated to thin film morphology is not well explained. Also no detailed discussions related with device performance. I would not suggest this is a topnotch research, and much more effort is needed.*

Response: We have now added more film morphology characterization and structure-property relationship discussion in the revised manuscript following reviewers' comments/suggestions. Please refer to the responses to the reviewer 1 and 2.

REVIEWERS' COMMENTS

Reviewer #1 (Remarks to the Author):

The authors have addressed most comments from the reviewers. I can recommend its publication after they address the below concerns.

- (1) Figure 3b should include the Reference 82 with 19% PCE.
- (2) Lots of literatures have reported that the non-covalent interaction plays an important role in molecule design. In this regard, the authors should further clarify the novelty in the manuscript.
- (3) Figure 3b is dated because there have been lots of references with PCE over 17%.
- (4) Could the authors give more solid proof and explanation to the more planar structure exist in thin film.
- (5) Could the authors employ a better processing method which have reported to offer a better performance as 16% efficiency is relatively low at this stage.

Reviewer #2 (Remarks to the Author):

The authors have revised the manuscript according to the suggestions and it can be accepted now.

Response to the referee reports regarding our manuscript entitled “*Tuning Polymer-Acceptor Backbone Coplanarity and Conformational Order to Achieve High-Performance Printed All-Polymer Solar Cells with Reduced Charge Recombination*”

Reviewer #1

Comments: *The authors have addressed most comments from the reviewers. I can recommend its publication after they address the below concerns.*

Response: We thank Referee #1 for their positive review. Below, we have outlined in detail the changes we made to our manuscript in order to address reviewer suggestions.

- 1) *Figure 3b should include the Reference 82 with 19% PCE.*

Response: We thank the reviewer for the suggestion. We have updated Figure 3b (renamed Fig. 4b) as suggested.

- 2) *Lots of literatures have reported that the non-covalent interaction plays an important role in molecule design. In this regard, the authors should further clarify the novelty in the manuscript.*

Response: There are relatively few reports on the non-covalent interaction effect in OPVs. Even less is the use of this principle in the all-PSCs. Importantly, we show that the non-covalent interaction effect significantly improves the active layer blend morphology and allows high performing blade coated all-PSC to be achieved. We believe that our results, and particularly the use of bithiazole based linker, will inspire more researcher on exploring in this direction. To further explain the novelty in the manuscript, we have added following statements in the introduction section: “Moreover, a high PCE of 14.7 (PCE_{ave} = 13.9 ± 0.6 %) can be achieved in devices fabricated using solution shearing under ambient conditions, and long-time stability of devices was observed. This PCE is among the highest for devices made under conditions relevant to scalable printing techniques. Our work shows that the unique geometric and electronic properties of BTz make it a promising building block for new classes of polymer acceptors and demonstrates that conjugated polymers with more planar and rigid backbone conformation through π -linker optimization and non-covalent bonding interactions can simultaneously achieve high electron mobility, good polymer-blend miscibility, and minimal performance losses during the transition to large-scale manufacturing of printed OPVs.”.

- 3) *Figure 3b is dated because there have been lots of references with PCE over 17%.*

Response: We thank the reviewer for the suggestion. We have added more recent references of high-performing binary All-PSCs in Figure 3b (renamed Fig. 4b). The references are tabulated in the Supplementary Information section F.

- 4) *Could the authors give more solid proof and explanation to the more planar structure exist in thin film.*

Response: Compared with PY-2T, PY-BTz is designed to have a more co-planar and rigid polymer backbone thanks to 1) the S(thiazolyl)···O(alkoxy) attraction promoting a non-covalent conformational lock effect and 2) the replacement of (thiophene)C–H with (thiazole)N eliminating repulsive C–H···H–C interactions with neighboring arene units. The computed torsional potential energy surfaces (Figure 1e-f) indicate that the presence of the non-covalent conformation lock in PY-BTz, combined with removal of repulsive (C-H)-(C-H) interactions, hinders bond rotation, suggesting that the molecular design proposed favors an overall more coplanar backbone for PY-BTz. These results are fully consistent with experimental XRD (closer π - π stacking distance), FTIR (Figure S2, shift of CN stretching toward higher energy) and UV-Vis measurements (a red-shifted electronic absorption on-set, increased resistance toward thermally induced UV-Vis absorption spectra change, and reduced Urbach energy). It should be noted that these structural and physical properties are in good agreement with the device electrical (i.e., increased electron mobility) and photophysical parameters (superior exciton dissociation and suppressed charge recombination), which further corroborates the more planar structure of PY-BTz in thin films.

- 5) *Could the authors employ a better processing method which have reported to offer a better performance as 16% efficiency is relatively low at this stage.*

Response: We thank the reviewer for making this suggestion. However, we believe the further processing optimization, which requires additional optoelectronic and morphological characterization, is beyond the scope of this work that has main focus on the molecular design, morphological characterization, and compatibility with blade-coating techniques.

Reviewer 2

Comments: *The authors have revised the manuscript according to the suggestions and it can be accepted now.*

Response: We thank this Reviewer for the positive feedback.